# FACTBENCH: A DYNAMIC BENCHMARK FOR IN-THE-WILD LANGUAGE MODEL FACTUALITY EVALUATION

## ABSTRACT

Language models (LMs) are widely used by an increasing number of users, underscoring the challenge of maintaining factual accuracy across a broad range of topics. We present VERIFY (**V**erification and **E**vidence **R**etr**I**eval for **F**actualit**Y** evaluation), a pipeline to evaluate LMs' factual accuracy in real-world user interactions. VERIFY considers the verifiability of LM-generated content and categorizes content units as `supported`, `unsupported`, or `undecidable` based on the retrieved web evidence. Importantly, VERIFY's factuality judgments correlate better with human evaluations than existing methods. Using VERIFY, we identify "hallucination prompts" across diverse topics–those eliciting the highest rates of incorrect or unverifiable LM responses. These prompts form FACT-BENCH, a dataset of 985 prompts across 213 fine-grained topics. Our dataset captures emerging factuality challenges in real-world LM interactions and is regularly updated with new prompts. We benchmark widely-used LMs from GPT, Gemini, and Llama3.1 family on FACTBENCH, yielding the following key findings: **(i)** Proprietary models exhibit better factuality, improving from `Hard` to `Easy` hallucination prompts. **(ii)** Llama3.1-405B-Instruct shows comparable or lower factual accuracy than Llama3.1-70B-Instruct across all evaluation methods due to its higher subjectivity that leads to more undecidable content. **(iii)** Gemini1.5-Pro shows a significantly higher refusal rate, with over-refusal in 25% of cases.

## 1 INTRODUCTION

Despite ongoing efforts to enhance their factuality, language models (LMs) continue to generate inaccurate or arbitrary content, known as hallucinations (Huang et al., 2023; Liu et al., 2023). The widespread use of LMs and the evolving nature of information demand a dynamic factuality evaluation benchmark to identify the challenges LMs face in real-world applications. Current long-form factuality evaluation benchmarks (Min et al., 2023; Wei et al., 2024b; Malaviya et al., 2024) are limited by their static nature and limited coverage of usage scenarios. The static design makes these benchmarks susceptible to overfitting and potential information leakage (Magar & Schwartz, 2022), rendering them unsuitable for capturing factuality challenges in daily LM usages. Moreover, existing benchmarks often focus on a limited subset of tasks. For instance, data used in developing FactScore Min et al. (2023) primarily addresses biographical queries, while ExpertQA (Malaviya et al., 2024) recruits human experts to curate domain-specific questions. Other benchmarks (Chen et al., 2023; Wei et al., 2024b) cover a limited number of domains and are either LM-generated or human-curated, further limiting their applicability to in-the-wild scenarios.

In this work, we make two primary contributions: **(i)** a dynamic and diverse factuality evaluation benchmark curated from real-world LM usage and **(ii)** a factuality evaluation framework that measures the appropriateness of in-the-wild prompts for inclusion in our benchmark by estimating how frequent strong LMs generate unverifiable and incorrect responses. Concretely, we introduce **FACT-BENCH**, an updatable benchmark grounded in the real-world usage of LMs. FACTBENCH comprises 985 diverse information-seeking prompts, across 213 topics, that tend to cause LMs to produce unverifiable and incorrect responses. Using clustering methods, we begin by identifying 382 unique tasks within the LMSYS-Chat-1M dataset Zheng et al. (2024). Prompts within each task cluster are then labeled for verifiability, indicating whether the prompt's response can be verified using Google search results. We assess the usefulness of prompts by considering factors such as clarity, inter-

est, and relevance to a broad audience. Finally, verifiable prompts that meet a specified usefulness threshold are considered to be included in FACTBENCH.

We further define the "hallucination prompts" as the ones that tend to elicit incorrect or unverifiable content from a selected group of strong LMs. To identify such prompts, we design **VERIFY**, a **V**erification and **E**vidence **R**etr**I**eval for **F**actualit**Y** evaluation pipeline to detect nonfactual content in LM responses. VERIFY first extracts content units from model responses and identifies their type. It then evaluates only the verifiable units against web-based evidence using an interactive query generation and evidence retrieval technique. Finally, VERIFY categorizes units as supported, unsupported, or undecidable based on the evidence. We also propose a hallucination score based on the number of unsupported and undecidable labels to quantify the degree of hallucination in model responses. This score helps measure the appropriateness of the corresponding user prompts for our final benchmark. To further refine our selection process, we categorize prompts into three tiers (Hard, Moderate, and Easy) based on the overall performance of the evaluated LMs. We then select prompts with the highest hallucination scores within each tier, resulting in a final benchmark of 985 prompts after manual inspection.

To assess the performance of widely-used LMs on FACTBENCH, we evaluate two proprietary, GPT4-o (OpenAI, 2024b), Gemini1.5-Pro (Team et al., 2024), and two open-weight models from Llama3.1 family, i.e., Llama3.1-70B-Instruct and Llama3.1-405B-Instruct (Meta, 2024), across all tiers of FACTBENCH. The results show that LM performance significantly increases across tiers, aligning with our curation strategy. To compare the effectiveness of different factuality evaluation methods, we use VERIFY units as a common basis and feed them into factuality evaluation baselines for verification. Our results reveal that VERIFY achieves the highest correlation with human judgments compared to state-of-the-art methods. This finding underscores the effectiveness of our approach in benchmark creation and factuality assessment.

In summary, our contributions are as follows:

- We introduce FACTBENCH, a new benchmark grounded in the real-world usage of LMs. FACTBENCH is designed to be adaptable by continuously incorporating newly collected hallucination prompts that cause LMs to generate incorrect content. This dynamic approach ensures that the benchmark remains relevant, addressing the evolving challenges in factual generation.
- We design VERIFY, a factuality evaluation pipeline that considers the verifiability of generated content and categorizes units into supported, unsupported, or undecidable according to retrieval results. VERIFY addresses the limitations of prior work that only makes binary decisions on supportedness and archives the highest average correlation with human evaluations.
- We release our human factuality annotations on 5,519 content units, with each unit independently evaluated by two annotators. Each annotator evaluates the independence of extracted units and their factuality using existing online resources. This human-annotated data provides quantifiable evaluation resources for assessing future factuality evaluation techniques.

## 2 RELATED WORK

### 2.1 FACTUALITY EVALUATION BENCHMARKS

The widespread adoption of LMs, coupled with their tendency to hallucinate, demands new benchmarks that can effectively identify their factual weaknesses across diverse scenarios.

Prior factuality evaluation benchmarks mainly focus on short-form and human-curated question answering (QA) tasks. For instance, TruthfulQA (Lin et al., 2022), HaluEval (Li et al., 2023), and FELM (Chen et al., 2023) mostly focus on short-form knowledge-based QA with questions with human-selected topics, despite LMs typically engaging in long-form conversations. The data used in developing FactScore (Min et al., 2023), while long-form, is limited to a single, fairly easy task of biographical QA. LongFact (Wei et al., 2024b) expands to 38 human-selected topics, but the prompts are LM-generated rather than user-driven. FactCheck-Bench (Wang et al., 2024a) collects ChatGPT hallucinations from Twitter, but its scope is narrow (94 prompts) and focuses on a specific and rather obsolete model. Moreover, all these datasets are static and prone to the data contamination issues (Magar & Schwartz, 2022). We fill these gaps by offering a benchmark that systematically mines hallucination prompts from in-the-wild user-model chat logs across various topics. FACTBENCH is

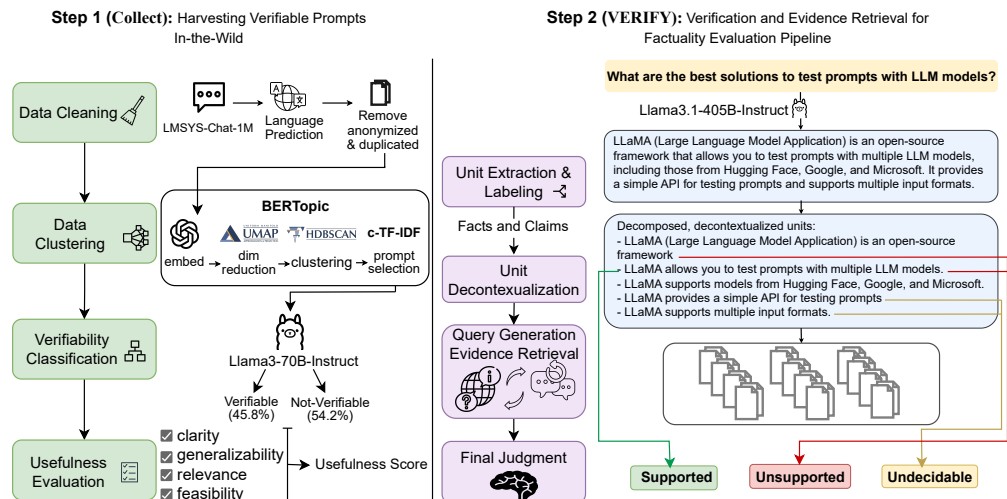

Figure 1: This figure outlines the two-step process we use to evaluate LM responses. Step 1 (left) involves cleaning, clustering, and evaluating prompts for verifiability and usefulness. Step 2 (right) decomposes responses into units, retrieves external evidence, and generates factuality labels (`supported`, `unsupported`, `undecidable`) with a hallucination score to flag inaccuracies. This process involves the collection and appropriateness assessment of hallucination prompts.

designed to be regularly updated as new prompts are gathered from real-world interactions, overcoming the constraints of fixed time frames and ensuring ongoing relevance to evolving LM capabilities and usage patterns.

## 2.2 FACTUALITY EVALUATION METHODS AND VERIFIABILITY

The challenge of distinguishing verifiable from non-verifiable claims is central to fact-checking. AFaCTA (Ni et al., 2024) stresses that claims are verifiable when they provide sufficient specificity for evidence retrieval. The subjective nature of check-worthiness, shaped by political and social contexts (Konstantinovskiy et al., 2020; Nakov et al., 2022), complicates this, particularly in LM-generated content where fact-opinion lines blur (Vosoughi et al., 2018). To address this, VERIFY introduces an `undecidable` label for claims with ambiguous factuality, accommodating both objective and context-dependent claims (see details in Section 4.1).

Long-form content evaluation presents unique challenges due to its complexity and the numerous claims it typically contains. To address these challenges, SAFE (Wei et al., 2024b) and FactScore (Wang et al., 2024b) decompose content into individual facts for granular verification. Our method, VERIFY, builds upon this approach by not only decomposing LM-generated content into units but also distinguishing between verifiable and non-verifiable elements that appear in user-model interactions. VERIFY focuses exclusively on verifiable units for further verification. It classifies these units as `supported` or `unsupported` only when confident evidence is found, labeling them `undecidable` otherwise. This approach introduces a more robust framework for evaluating the factual precision of LM-generated content. In contrast, Factcheck-GPT extracts coarser-grained units and heavily relies on parametric knowledge, limiting its reliability for factual evaluation.

## 3 HARVESTING HALLUCINATION PROMPTS FROM LM INTERACTIONS

Our current understanding of LM performance on verifiable tasks is limited, and existing factuality evaluation benchmarks cover only a narrow range of verifiable use cases.

To address this gap, we collect English prompts from the first turn of conversations in the LMSYS-Chat-1M dataset (Zheng et al., 2024), which is a large-scale, in-the-wild LM conversations dataset. Our objective is to identify prompts that are verifiable, diverse, and relevant for evaluating model responses through a multi-step process. Figure 1 (left) outlines our collection process.

- **Data Clustering**: After cleaning the data (see details in Appendix 11.1), we get 294,333 distinct prompts and cluster them into various topics. We use BERTopic (Grootendorst, 2022), a dynamic topic-modeling pipeline that (1) embeds prompts using OpenAI's text-embedding-3-small model (OpenAI, 2024a), (2) applies UMAP for dimensionality reduction, and (3) employs HDBSCAN, a hierarchical density-based clustering algorithm. HDBSCAN identifies 142,702 (48.5%) of the prompts as outliers, which we exclude to remove overly-specific prompts. Finally, we use a class-based TF-IDF method to select the top 100 most representative prompts from each cluster and summarize them into concise topics (up to 10 words) using GPT-4 Turbo (OpenAI, 2024c).

- **Verifiability Classification**: We focus on prompts that ask for varying degrees of verifiable responses. To identify these, we employ Llama3-70B-Instruct (AI@Meta, 2024) to distinguish between verifiable and non-verifiable prompts (see classification prompt and the portion of verifiable prompts within each cluster in Appendix 11.9.2 and Figure 7). Overall, verifiable prompts constitute 45.8% of total prompts from the previous step.

- **Usefulness Evaluation.** The remaining collection contains around 70K prompts, too large for manual or automated fact-checking Randomly selecting a subset for evaluation would be suboptimal, as it may include unclear or over-specific requests. Therefore, we propose a set of criteria to identify *useful* prompts. A useful prompt needs to be (**i**) clear and understandable, (**ii**) generalizable to various users or scenarios, (**iii**) has potential interest or value to a broader audience, and (**iv**) is within the capabilities of LMs (e.g., excludes real-time data). To reduce single-model bias, we use two language models, GPT-4-Turbo and Llama3-70B-Instruct, as annotators. Each model scores prompts on a scale of 1 (low) to 5 (high) for each criterion. The scoring prompt is provided in Appendix 11.9.3. The final usefulness score for each prompt is calculated as the average score across all criteria, summing the score from two models. The usefulness scores are then used to select an initial set of prompts, as we describe in Section 5.

## 4 VERIFY: Verification and Evidence Retrieval for Factuality Evaluation

In this section, we design an automatic factuality evaluation pipeline called VERIFY to measure the *appropriateness* of these prompts, i.e., based on whether they tend to elicit unfactual responses from LMs, for inclusion into the FACTBENCH. Our goal is to verify model responses in natural user-LM settings, where responses may include a mixture of verifiable and non-verifiable statements. To evaluate model responses efficiently, we first establish the key criteria for determining the verifiability of statements.

### 4.1 Factual Evaluation through the Lens of Verifiability

A statement is verifiable if it provides sufficient information to guide fact-checkers in verification (Ni et al., 2024). We classify *verifiable statements* into two categories:

**Context-independent Statements**: These are objective assertions that can be directly verified against knowledge sources. For example, "RTX 3060 has a memory bandwidth of 360 Gbps.".

**Context-dependent Statements**: These statements require additional information for verification. For instance, to verify the statement "The difference in memory bandwidth between the RTX 3060 and RTX 3060 Ti is relatively small." one needs knowledge of both GPUs' bandwidths and an understanding of what qualifies as *relatively small* in this context.

By focusing only on verifiable statements during user-LM interactions, we can assess the factuality of the response in a more efficient and accurate manner, as we will describe next.

### 4.2 Unit Extraction and Labeling

User requests span a wide range of domains (see the Figure in 2), and model responses contain a variety of content types. To evaluate verifiable statements, we first decompose the model response into independent content units. A content unit can represent a `Fact`, a `Claim`, an `Instruction`, a `Disclaimer`, a `Question`, or other relevant content that appears during user-model interaction. Many of these content units are not verifiable, meaning they cannot be fact-checked. Examples

include questions and disclaimers, which often pertain to the context of the conversation or the capabilities of the model itself rather than presenting factual information.

We label each unit by its type to identify those suitable for verification. We use the Llama3-70B-instruct model, which serves as the backbone LM for all tasks in this pipeline, to extract and label content units. A carefully crafted prompt with examples (see Appendix 11.9.4) guides the model in this task. We classify objective statements as `Fact` and potentially subjective, context-dependent statements as `Claim`. Only units labeled as `Fact` or `Claim` are passed to next step.

### 4.3 Unit Decontextualization

Gunjal & Durrett (2024) highlights the importance of "molecular units"—units that contain sufficient information to be uniquely identifiable in factuality assessment. Inspired by that, we implement a unit decontextualization component in our pipeline to minimally revise verifiable units and make them self-contained. The prompt is provided in Appendix 11.9.5.

### 4.4 Query Generation and Evidence Retrieval

In order to verify the self-contained units, we need to retrieve relevant evidence from knowledge sources. We utilize SerperAPI[1] for Google Search and web-evidence retrieval.

To enhance search quality and ensure the retrieval of relevant evidence that best assists in verification, we implement an interactive query refinement technique. The query generator operates iteratively within an interactive feedback loop. It first produces a query for the target unit, which is then used in Google Search to return relevant snippets. In subsequent rounds, the query generator evaluates the relevance of the retrieved snippets for verifying the target unit and refines the query accordingly. This iterative process continues for up to five rounds, progressively improving the query's quality and relevance. The final set of queries and associated search results are then passed to the next step for final judgment. The prompt is provided in Appendix 11.9.6.

### 4.5 Final Answer Generation

In this step, the model (Llama3-70B-Instruct) is tasked with making a final decision on the units' accuracy by evaluating multiple rounds of retrieved evidence using Chain-of-Thought prompting Wei et al. (2024a). For each unit, the model **(i)** summarizes the relevant knowledge points, **(ii)** assesses their relationship to the unit, and **(iii)** determines whether the evidence supports (`supported`), refutes (`unsupported`), or is inconclusive (`undecidable`). The prompt is provided in Appendix 11.9.7. This results in annotation labels for all verifiable units in the original model response. An overview of our evaluation pipeline is provided in the right part of Figure 1.

### 4.6 Hallucination Score

After annotating the individual content units, we propose a hallucination metric to quantify the incorrect and undecidable contents within a model's response, $R_M$. Let $U$ represent the set of `undecidable` units, $C$ the set of `unsupported` units, and $V$ the total set of verifiable units (`Claims` and `Facts`). The final *Hallucination Score* is computed as follows:

$$H(R) = \frac{|C| + \alpha|U|}{\sqrt{|V|}} \tag{1}$$

Here, $\alpha \in (0, 1)$ is a weighting factor that controls the relative importance of `undecidable` units compared to `unsupported` ones. This adjustment is necessary as `undecidable` units may be correct but lack sufficient evidence for verification, or the VERIFY pipeline might be unable to verify them. For experiments, we set $\alpha = 0.5$. The denominator, $\sqrt{|V|}$ ensures that the weight of errors (`unsupported` and `undecidable` units) grows more significantly relative to the total number of verifiable units, placing stronger emphasis on even a few errors without allowing large sets of units to mask their importance. Using this score, we rank collected prompts to curate a benchmark, as described in the following section.

---

[1]https://serper.dev/

| Benchmark | In-the-Wild | Dynamic | # Prompts |
|---|---|---|---|
| FELM (Chen et al., 2023) | ✗ | ✗ | 847 |
| ExpertQA (Malaviya et al., 2024) | ✗ | ✗ | 484 |
| FactScore (Min et al., 2023) | ✗ | ✗ | 500 |
| LongFact (Wei et al., 2024b) | ✗ | ✗ | 2280 |
| FactCheckBench Wang et al. (2024a) | mixed | ✗ | 94 |
| **FACTBENCH** | ✓ | ✓ | 985 |

**Common Factual Requests**

**6.2%** Travel itineraries
**3.9%** Recipe requests
**3.7%** Medical questions
**2.2%** LM capabilities
**1.9%** LM apps (i.e. in education)
**1.8%** GPU recommendations
**1.5%** Game comparisons
**1.5%** Theoretical concepts
**1.4%** Solar inquiries
**1.4%** Music recommendations

Figure 2: Statistics of different factuality benchmarks. FACTBENCH is the first dynamic and in-the-wild factuality evaluation benchmark with diverse topic coverage.

## 5 FACTBENCH DATASET

### 5.1 THREE TIERS OF PROMPTS

Using the Hallucination score, we can measure the appropriateness of the collected prompts for inclusion in our final dataset. However, weaker LMs have a higher tendency to hallucinate. To prevent prompts queried to such LMs from populating the whole dataset, we categorize the prompts into three tiers: Tier 1: `Hard`, Tier 2: `Moderate`, and Tier 3: `Easy` based on the overall performance of the model used to generate responses. We rely on the LM ranking provided by the Chatbot Arena Leaderboard website[2] to do this categorization. Models and the statistics of each tier can be found in Appendix Table 5. Heuristically, we selected prompts with a usefulness score of 4 or higher from `Hard`, 4.5 or higher from `Moderate` and exactly 5 from `Easy`. This approach assumes that `Hard` responses can best approximate prompt appropriateness, allowing for a lower usefulness threshold to capture more prompts, with higher standards for subsequent tiers. Our initial prompt set includes 4.2K prompts, 53% from `Hard`, 34% from `Moderate`, and 13% from `Easy`.

### 5.2 BENCHMARK CURATION

We measure prompt appropriateness in each tier using the hallucination score (Equation 1) of their corresponding LM responses. We select 1.1K prompts with the highest scores, maintaining the original tier proportions (64% `Hard`, 32% `Moderate`, 14% `Easy`). After manual inspection, we excluded 115 instances (details in Appendix 11.2), resulting in a benchmark of 985 prompts called FACTBENCH. Table 2 illustrates the benchmark statistics and compares it to other long-form factuality evaluation benchmarks. Our work introduces the first dynamic, real-world factuality evaluation benchmark comprising hallucination prompts across diverse topics.

## 6 EXPERIMENTAL SETUP

**Language Models**: We benchmark LMs against FACTBENCH to evaluate their performance on this dataset using different factuality evaluation methods. We evaluate the most recent and powerful models (available via APIs) from proprietary and open-source categories. From the proprietary models, we benchmark GPT-4o (omni) (OpenAI, 2024b) and Gemini1.5-Pro (Team et al., 2024). For open-source models, we evaluate the highest-ranked open-source models on Chatbot Arena Leaderboard[3], Llama3.1-70B-Instruct and Llama3.1-405B-Instruct (Meta, 2024).

**Baselines**: For comparison, we consider three reference-dependent factuality evaluation techniques: FactScore (Min et al., 2023), Search-Augmented Factuality Evaluator (SAFE) (Wei et al., 2024b), and Factcheck-GPT (Wang et al., 2024a). A detailed description of these methods and their experimental setup is provided in Appendix 11.3.

---

[2]Chatbot Arena Leaderboard provides an open platform that ranks LMs performance through pairwise human comparison.

[3]https://huggingface.co/spaces/lmsys/chatbot-arena-leaderboard

Table 1: Results of VERIFY and baselines across three tiers of FactBench and 4 widely-used LMs using the factual precision score (Equation 3), as in prior work. For each evaluation method and within each tier, the best and second-best factuality scores are highlighted in blue and green, respectively. Proprietary models are more factual, with factuality improving from Hard to Easy prompts.

| FactBench | Model | FactScore | SAFE | Factcheck-GPT | VERIFY |
|---|---|---|---|---|---|
| Tier 1: `Hard` | GPT4-o | 53.19 | **63.31** | **86.40** | **71.58** |
| | Gemini1.5-Pro | 51.79 | 61.24 | 83.45 | 69.38 |
| | Llama3.1-70B-Instruct | 52.49 | 61.29 | 83.48 | 67.27 |
| | Llama3.1-405B-Instruct | **53.22** | 61.63 | 83.57 | 64.94 |
| Tier 2: `Moderate` | GPT4-o | **54.76** | **65.01** | **89.39** | **76.02** |
| | Gemini1.5-Pro | 52.62 | 62.68 | 87.44 | 74.24 |
| | Llama3.1-70B-Instruct | 52.53 | 62.64 | 85.16 | 72.01 |
| | Llama3.1-405B-Instruct | 53.48 | 63.29 | 86.37 | 70.25 |
| Tier 3: `Easy` | GPT4-o | 69.44 | 76.17 | **94.25** | **90.58** |
| | Gemini1.5-Pro | 66.05 | 75.69 | 91.09 | 87.82 |
| | Llama3.1-70B-Instruct | 69.85 | **77.55** | 92.89 | 86.63 |
| | Llama3.1-405B-Instruct | **70.04** | 77.01 | 93.64 | 85.79 |

# 7 EXPERIMENTAL RESULTS AND FURTHER ANALYSES

## 7.1 FACTUALITY IMPROVES WITH PROPERIARY LMS AND EASIER PROMPTS

To compare model performance on FACTBENCH, we use the factual precision metric proposed by Min et al. (2023). This metric quantifies an LM's factuality by calculating the proportion of `supported` units among all extracted units in a response, averaged across all responses (detailed in Appendix 11.4).

Table 1 compares the factual precision of LMs on FACTBENCH as measured by different evaluation methods. Despite all baselines verifying responses at the finest granularity (standalone units), we observe varying factual precision ranges across different pipelines. We observe that VERIFY consistently maintains the same ranking of models across all three tiers, unlike other evaluation methods. VERIFY ranks **GPT4-o as having the highest factual precision across all tiers, followed by Gemini1.5-Pro and the two open-source Llama3.1 models**. This consistency suggests that VERIFY may provide a more stable and reliable evaluation of model performance across varying difficulty levels.

Surprisingly, Llama3.1-405B-Instruct demonstrates comparable or inferior factuality to Llama3.1-70B-Instruct according to VERIFY. Further investigation (Figure 4) reveals that Llama3.1-405B-Instruct has a lower proportion of `unsupported` units but the highest proportion of `undecidable` units across all LMs. This primarily stems from stronger subjectivity in Llama3.1-405B-Instruct, with more frequent use of subjective adjectives such as "solid," "exclusive," and "well-known." The rigorous reasoning logic enforced in our pipeline typically classifies these subjective elements as `undecidable`, thereby reducing the factual precision. Detailed analysis and examples are provided in Appendix 11.8.

Another significant observation is **the consistent improvement in factuality precision across LMs from the `Hard` to `Easy` tiers**, as shown by all evaluation methods. This aligns with our benchmark curation approach, where hallucination prompts are ranked based on their tendency to provoke incorrect or unverifiable responses. `Easy` prompts are less likely to trigger hallucinations in strong models, as their appropriateness is measured based on weaker LM hallucinations.

## 7.2 ON FACTUALITY EVALUATION VERIFY STRONGLY CORRELATES WITH HUMAN

The factuality of a model, measure by a factuality evaluation method, depends on the granularity of the extracted units and the method's verification capabilities. FactScore extracts units with a finer granularity compared to VERIFY due to its focus on biographical text evaluation, which typically consists of objective and easily separable units. On the other hand, Factcheck-GPT's claim-level

Table 2: Response-level correlation between factuality evaluation methods and human annotations of 50 prompts across 4 LMs. **F** refers to `Factual` labels, and **O** refers to `Other`. **Independent Units** are independent units identified by both human annotator, and **All Units** include both dependent and independent units. VERIFY achieves the highest correlation with human judgments among factuality evaluation methods (highlighted in green ).

| | Independent Units | | | | All Units | | | |
|---|---|---|---|---|---|---|---|---|
| | Factcheck-GPT | FactScore | SAFE | VERIFY | Factcheck-GPT | FactScore | SAFE | VERIFY |
| **Pearson (F/O)** | 0.90 / 0.64 | 0.86 / 0.54 | 0.88 / 0.60 | **0.90 / 0.65** | 0.96 / 0.62 | 0.92 / 0.35 | 0.95 / 0.59 | **0.97 / 0.71** |
| **Spearman (F/O)** | 0.87 / 0.63 | 0.87 / 0.61 | 0.87 / 0.63 | **0.89 / 0.70** | 0.94 / 0.58 | 0.90 / 0.42 | 0.93 / 0.53 | **0.95 / 0.68** |

decomposition (finest-level) often results in sentence-level containing multiple claims. To establish a unified evaluation framework for these methods, we selected 50 FACTBENCH prompts from 50 randomly sampled topics (one prompt per topic) and generated responses from each model. We then applied our unit extraction (Section 4.2) and decontextualization approach (Section 4.3) to decompose generated LM responses into *self-contained* and *verifiable* units. This method was chosen for its ability to handle user-model conversations (with careful instructions and in-the-wild demonstrations), extract moderately granular units, and filter them based on verifiability.

Three fluent English speakers are hired to annotate a total of 200 LM responses for 4 models on the same set of 50 prompts. VERIFY breaks LM responses into 5,519 units, with each response annotated by two annotators. Each annotator evaluated the independence and factuality of each unit. A unit is considered *Independent* if it is *verifiable* and *self-contained*. A *Dependent* unit, on the other hand, is either an unverifiable piece of information, e.g., "I can provide you with some examples." or under-specified content. For example, "She won the best actress award" is underspecified as it contains a pronoun, and it's unclear what specific award is being referenced. Overall, 80.9% units are considered *Independent* by both annotators with an inter-annotator agreement of $0.52$ in terms of Cohen's Kappa score. Additionally, each annotator evaluates the factuality of a unit by using two labels: (1) *Factual* when supporting evidence can be found using web search and (2) *Other* when refuting evidence is found, or the factuality cannot be determined. The indecisiveness usually happens when the unit is dependent. Indecisiveness typically occurs with *Dependent* units. Annotators agree on the factuality of 84.8% of units, with a Cohen's Kappa agreement of 0.55. These binary factuality labels are compatible with VERIFY and other baselines. A unit is *independent* if both annotators agree; otherwise, it is labeled dependent. Factuality is decided in the same manner.

### 7.2.1 ACCURACY DOES NOT CAPTURE ALIGNMENT

In our first experiment, we fed only the independent units, as agreed by both annotators, to the factuality evaluation methods for verification. Figure 3 compares the accuracy of different methods against human labels. While VERIFY demonstrates superior performance on LLaMA3.1-405B-Instruct compared to other baselines, it is generally outperformed by Factcheck-GPT. Further analysis reveals that Factcheck-GPT achieves the highest average precision in predicting factual units (85.2% compared to VERIFY's 75.4%). This performance gap is attributed to the distinct design choices of the two methods. Factcheck-GPT leverages its internal knowledge when no external evidence is found, whereas VERIFY heavily relies on external knowledge to make decision-making and labels units as `undecidable` without evidence. Although VERIFY's approach may result in lower factuality, it maintains high confidence and trustworthiness in its evaluations, particularly in cases where conservative, evidence-based decisions are preferable. Therefore, to better capture the human-method decision alignment, we calculate Pearson and Spearman correlations following previous work (Wei et al., 2024b; Min et al., 2023). As shown in the left part of Table 2 (Independent Units), **VERIFY consistently outperforms other methods using correlation estimation techniques**. Notably, VERIFY achieves a significantly higher correlation with human annotation in the *Other* category. This highlights our method's nuanced handling of `undecidable` cases and its close alignment with human judgment.

### 7.2.2 VERIFY IS ROBUST TO AMBIGUOUS INPUTS

Previously, we evaluated the ability of different methods to verify independent units. However, factuality evaluation methods often generate both dependent and independent units, requiring mech-

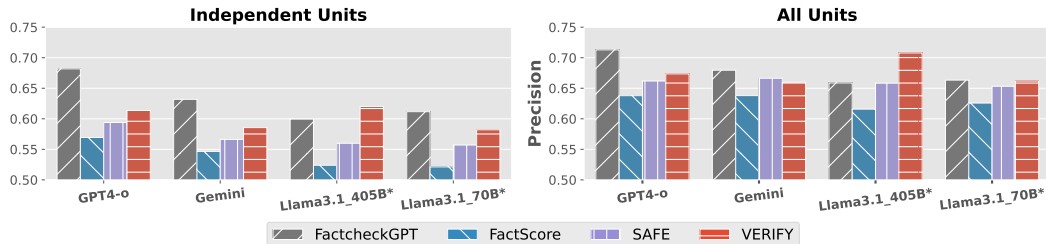

Figure 3: Accuracy of factuality evaluation method predictions compared to human annotations across LMs (*Instruct version). **Independent Units** are independent units identified by both human annotators and **All Units** include both dependent and independent units. VERIFY is robust to dependent units, as demonstrated by the noticeable accuracy improvement in the right figure.

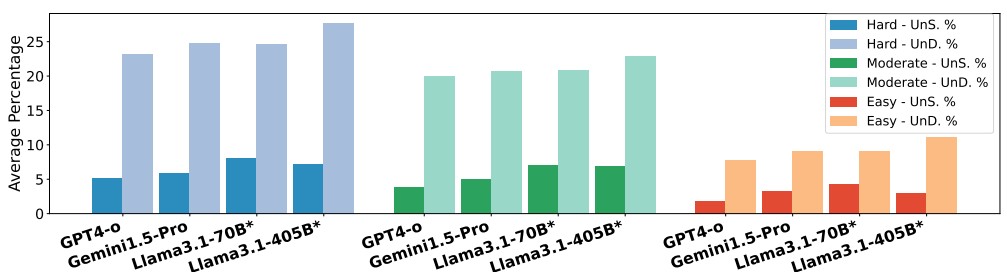

Figure 4: Average Percentage of `unsupported` (UnS) and `undecidable` (UnD) labels in different LMs (*Instruct version) evaluated by VERIFY. Llama3.1-405B-Instruct has the highest proportion of `undecidable` units across all LMs.

anisms to handle ambiguous units that are rather difficult to verify using existing online knowledge. In this experiment, we evaluate the robustness of factuality evaluation methods when handling ambiguous and contextually dependent units. As shown in the right part of Figure 3, the gap between VERIFY and the best-performing baseline, Factcheck-GPT, narrows to an average accuracy difference of 0.2%. This suggests 's improved accuracy relative to other baselines when dealing with dependent units. The correlation results, illustrated in the right part of Table 2 (All Units), further support this finding. **VERIFY consistently demonstrates the highest correlation with human judgments, indicating our method's nuanced handling of `undecidable` cases**. Our approach better reflects the reality that not every verifiable unit can be confidently judged and aligns more closely with human reasoning in ambiguous scenarios.

### 7.3 GEMINI1.5-PRO'S REFUSAL RATE IMPACTS ITS FACTUALITY

Current factual precision metrics (detailed in 11.4) do not evaluate LM factuality when models refrain from answering. In this section, we investigate LMs' refusal rate and its significance.

Previous work (Min et al., 2023) relied on simple heuristics or keywords to identify refusal responses, but our initial investigation finds these methods unreliable. To address this, we prompt GPT-4 Turbo to detect refusals and categorize them into different types, including clarification requests, lack of knowledge, safety concerns, and misinformation risks. Figure 5 shows the refusal rates for each model across different FACTBENCH tiers (see Appendix 11.5 for the task prompt and distribution of refusal categories for different LMs). Notably, Gemini1.5-Pro exhibits a significantly higher refusal rate than other models, approaching 10% on the `Hard` portion of FACTBENCH. As shown, LMs refuse to answer prompts in the `Hard` tier more frequently, as it contains the most challenging hallucination prompts. Moreover, while Gemini1.5-Pro's refusal strategies can help prevent hallucinations (see example 3), the high rate of refusals impacts its overall factuality precision. Additionally, our manual inspection reveals that in 25% refusal cases, Gemini1.5-Pro's reason for not answering is not valid. For example, the model interpreted "give me studies on the recommended interval between COVID vaccines" as a request for medical advice and refused to answer. These

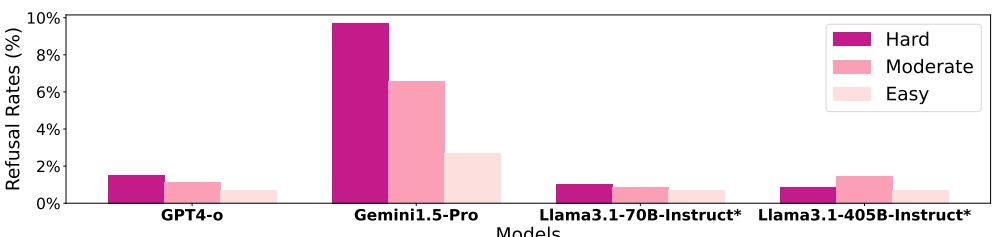

Figure 5: Refusal rate of different LMs across `Hard`, `Moderate`, and `Easy` tiers of FACTBENCH. Gemini1.5-Pro shows a significantly higher refusal rate than other LMs.

findings highlight the need for more nuanced factuality evaluation metrics to account for refusals as an important direction for future research.

## 8 CONCLUSION

We presented VERIFY, a factuality evaluation pipeline that automatically annotates the factuality of LM responses in real-world settings. VERIFY breaks down a response into content units, identifies verifiable ones, verifies them using retrieved web evidence, and categorizes each unit as `supported`, `unsupported`, or `undecidable`. Our method correlates strongly with human evaluations compared to existing state-of-the-art methods and is more robust in handling flawed units. Using VERIFY, we curated FACTBENCH, a benchmark of 985 prompts across 213 fine-grained topics that are grouped into three tiers: `Hard`, `Moderate`, and `Easy`. Prompts in each tier are sorted by appropriateness, i.e. their tendency to elicit hallucinated content from LMs. FACT-BENCH is the first real-world and diverse factuality evaluation benchmark designed to be regularly updatable to capture evolving factuality challenges faced by language models. Finally, we used our dataset to evaluate frontier LMs in proprietary and open-source categories. Our findings show that the proprietary model exhibits better factually, improving factuality from `Hard` to `Easy` tiers. Interestingly, Llama3.1-405B-Instruct shows lower factuality mainly due to its frequent use of subjective terms that are hard to verify.

Current factuality-evaluating metrics oversimplify the evaluation and do not consider model refusals and low factual recall, which is an important direction for future research in this field. Additionally, future work can explore more sophisticated evaluation pipelines that account for individual factual support as well as logical inter-unit connections. Such approaches would verify not only the factual accuracy of each content unit but also the logical coherence within the overall response, further improving the reliability of LM evaluation methods.

## 9 ETHICS STATEMENT

This research adheres to strict ethical standards, especially considering the potential impact of releasing datasets and methodologies designed for evaluating the factual accuracy of language models. All crowdsourced data used in this work was collected in accordance with relevant ethical guidelines, ensuring that participant privacy is respected and no personally identifiable information is included in the dataset. Specifically, the user-LM interaction dataset, LMSYS-Chat-1M, on which this research is based, has had all personally identifiable information removed, and we further filtered conversations potentially related to personal information during data cleaning. The potential risks associated with misusing our findings have been carefully considered. We aim to mitigate these risks by clearly delineating the scope and limitations of our models, ensuring that our dataset and methodology are used responsibly. Moreover, we acknowledge the possibility of inherent biases in crowdsourced data and have employed techniques to identify and minimize these biases in our models and evaluations. There are no conflicts of interest or external sponsorships that have influenced the outcomes of this research. All findings, including any potential model limitations, have been reported transparently to promote fairness, accuracy, and integrity in factuality evaluation tasks.

## 10 REPRODUCIBILITY STATEMENT

We have made significant efforts to ensure the reproducibility of the results presented in this paper. All experimental details, including the dataset collection and processing steps, are provided in Section 3 and Section 5 and further elaborated in Appendix 11.1. The proposed factuality evaluation framework, VERIFY, and its implementation are thoroughly described in Section 4, with additional details, including the prompts used and evaluation pipeline, in Appendices 11.9.4 through 11.9.7. To facilitate the reproducibility of our results, we will release the FACTBENCH and provide an anonymous link to the source code in the supplementary materials, enabling other researchers to replicate our benchmarks and use the evaluation metrics described in Section 7. Furthermore, we include all theoretical proofs and detailed statistical methods in the appendices, ensuring that all claims can be validated through the provided materials.

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

## 11 APPENDIX

### 11.1 DATA CLEANING

We begin by collecting prompts from the first turn of conversations in the LMSYS-Chat-1M dataset, which is a large-scale, in-the-wild LM conversations dataset. Since the existing language labels are unreliable, we employ the Llama3-70B-Instruct model (AI@Meta, 2024) to identify the language of each conversation using the prompt in Appendix 11.9.1. This gives us 516,771 distinct English prompts with at least 32 characters. Next, we remove anonymized (30.9%) and duplicated (12.1%) prompts. Meanwhile, we observed that some users queried LMs with thousands of identical prompts. To mitigate this issue's impact on subsequent clusters, we filter out prompts with a Jaccard similarity score greater than 0.9. Our cleaned data contains 294,333 distinct prompts.

### 11.2 MANUAL CHECK ON PROMPT VERIFIABILITY

In order to ensure the verifiability specified in Section 3, three authors have conducted multiple rounds of human inspection and validation to exclude all non-verifiable prompts like problem-solving (e.g., "A suit manufacturer has 14 suits for men and 4 suits for women. How many suits are available overall?") and faithfulness-related (e.g., "Translate the given text") tasks. More unverifiable examples are available in our prompt at Appendix 11.9.2. We excluded 115 prompts in the manual check process.

### 11.3 BASELINE DESCRIPTION

We use `gpt-3.5-turbo-0613` (Brown et al., 2020) as a backbone LM when running all baselines.

- **FactScore** (Min et al., 2023): FactScore evaluates the factual precision of LMs by breaking text into atomic facts and assessing the percentage of facts supported by Wikipedia articles. The original FactScore method is provided with Wikipedia pages with relevant information. However, the extracted units from in-the-wild requests are not associated with a Wikipedia page and might not even be found in Wikipedia articles. To make a fair comparison, we use the Wikipedia API (Goldsmith, 2014) to map these atomic units to the 5 closest Wikipedia topics in the Wiki database for retrieval.

- **Search-Augmented Factuality Evaluator (SAFE)** (Wei et al., 2024b): SAFE evaluates long-form factuality by decomposing text into atomic facts adopting the same FactScore fact extraction component and checking each fact's relevancy to the original query. For relevant facts, SAFE queries Google search engine for evidence retrieval, and label each fact as either supported or refuted accordingly.

- **Factcheck-GPT** (Wang et al., 2024a): Factcheck-GPT is a hallucination detection and mitigation framework. In the annotation phase, it assesses the factuality of LM-generated content using a multi-step annotation pipeline that includes the decomposition of claims, decontextualization, evidence retrieval through Google Search, evidence snippets generation, final factuality decision, and revision of non-factual elements. For this study, the final revision step is excluded from the baseline methodology.

### 11.4 FACTUAL PRECISION METRIC

We adopt the factual precision utilized by FactScore (Min et al., 2023) to compare the performance of different models on FACTBENCH. Given the set of prompts $P$ and knowledge source $K$, we first obtain model $M$ responses $\{R_M = M(p) \text{ for } p \in P\}$. All baselines decompose each response into atomic units (facts). Therefore, we denote $U$ to be the set of units in $R_M$. We calculate the **factual precision** of $R_M$ as:

$$f(R_M) = \frac{1}{|U|} \sum_{u \in U} \mathbb{I}[u \text{ is supported by } K] \qquad (2)$$

The overall factuality precision of each model on $P$ prompts is calculated as:

$$F(M) = \mathbb{E}_{p \in P}[f(M_p | M_p \text{ responds}] \qquad (3)$$

We do not consider factual recall; for instance, a model that abstains from answering too often or generates responses with minimal factual content. Wei et al. (2024b) suggests that there is a fixed number of content units users care about and that this number can be tuned. However, we did not find this metric compelling, as different models show different verbosity levels, making it difficult to establish a unified threshold. Moreover, the number of units is not a reliable indicator of quality, as the content may still lack relevance or usefulness.

## 11.5 REFUSAL PROMPT AND REFUSAL TYPE DISTRIBUTIONS

The refusal categories explain various reasons for declining to answer queries. "No Refusal" indicates a complete response, while categories like "Safety Concerns" and "Misinformation Risks" reflect avoidance of harmful or misleading information. Refusals may also stem from requests for "Sensitive or Private Information," where personal data is involved, or a "Clarification Request," where the model seeks further details. Other reasons include "Ethical and Legal Advice," "Hate Speech or Discrimination," and "Lack of Knowledge/Capability," which acknowledges the model's limitations. The "Other" category covers refusals that don't fit these specific reasons.

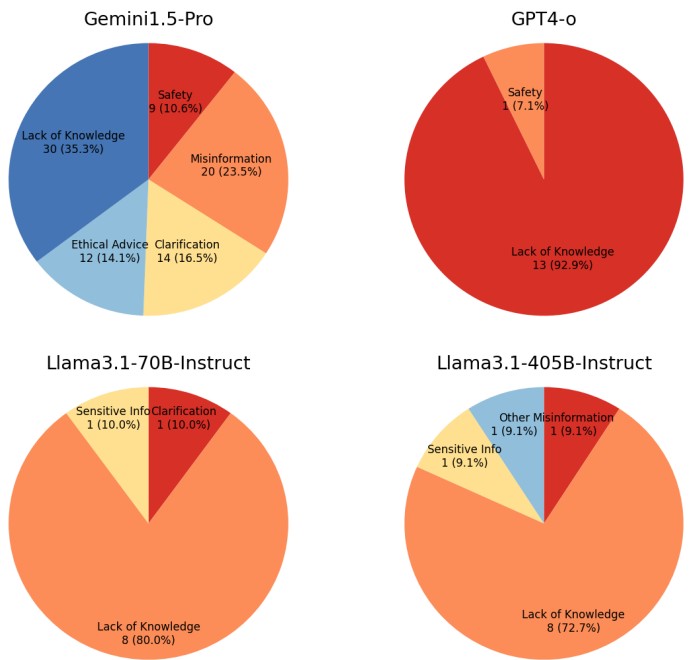

Figure 6: Distribution of LM's answer refusal categories on FACTBENCH

Figure 6 shows the distribution of refusal categories across LMs, with Gemini1.5-Pro having the highest number of refusals across a broad range of categories. We also show an example in Table 3 where Gemini105-Pro refrained from answering by requesting clarifications while GPT4-o produced a fully hallucinated response.

Table 3: Shows the varying behaviour of LMs when confronted with knowledge limitations. GPT4-o, Llama3.1-70B, Llama3.1-405B and produce hallucinated or inaccurate content (marked in red), Gemini1.5Pro either refrains from generating a response or requests additional clarification to better define the query.

| Model | Model Response |
|-------|----------------|
| GPT-4-o | Sure! Fuzhou, the capital of China's Fujian province, is known for its rich culinary culture. Here are some highly recommended restaurants where you can try a variety of delicious dishes:
1. **Ding Feng Zhuang**:
- *Cuisine*: Traditional Fujian ...
2. **Huangxiang Shousibao**:
- *Cuisine*: Seafood ...
3. **Yu Xiangyan**:
- *Cuisine*: Fujian and Cantonese...
4. **Juchunyuan** :
- *Cuisine*: Fujian...
5. **Shawanrong**:
- *Cuisine*: Modern Fusion... |
| Gemini | I can definitely recommend some restaurants in Fuzhou City! To give you the best suggestions, please tell me:
- What kind of cuisine are you interested in? (e.g., Fuzhou specialties, Cantonese, Sichuan, Western, etc.)
- What's your budget like? (e.g., fine dining, mid-range, budget-friendly)
- What's your dining style? (e.g., street food, casual, formal, romantic)
- What area of Fuzhou are you interested in? (Specific neighborhoods or landmarks will help!)
Once I have this information, I can provide you with a more tailored and helpful list of restaurant recommendations. |
| Llama 3.1 (70B) | Fuzhou, the capital of Fujian Province in China, is known for its rich culinary heritage and delicious local dishes. Here are some recommended restaurants in Fuzhou City:
1. **Buddha Jumps Over the Wall**: This restaurant is famous for its signature dish, "Buddha Jumps Over the Wall," a traditional Fujianese soup ...
2. **Fuzhou Fish Ball Restaurant**: This casual eatery is a local favorite for its delicious fish balls, which are made from fresh fish and served in a light broth.
3. **Gongdelin Vegetarian Restaurant**: This vegetarian restaurant offers a wide range of plant-based dishes, including traditional Fujianese options like stir-fried vegetables and tofu... |
| Llama 3.1 (405B) | Fuzhou, the capital of Fujian Province in China, is renowned for its rich culinary heritage... Here are some recommended restaurants in Fuzhou that showcase the best of Fujian cuisine and more:
1. **Dongbai Lou** - Known for its traditional Fujian dishes, including the famous Dongbai chicken and a variety of seafood options. ...
2. **Wu Ming Lou** - Offers a mix of Fujian and Cantonese dishes ...
3. **Fuzhou Wanshou Road Food Street** - While not a single restaurant, this food street is a must-visit for any food enthusiast...
4. **Min Jiang Lou** - Specializes in Fujian cuisine with a focus on river and sea delicacies...
5. **Huangcheng Hotel Restaurant** - Offers a range of Fujian dishes in a comfortable setting... |

Below is the prompt used for refusal detection and categorization:

```
Evaluate whether the following RESPONSE from a large language model contains a refusal to answer and
    categorize the type of refusal.

0. No Refusal: The response does not contain a refusal.
1. Safety Concerns: The response avoids answering due to harmful, dangerous, or illegal content in the
    query.
2. Misinformation Risks: The model refuses because answering might spread false or misleading
    information, especially in critical areas (e.g., medical, legal, or financial advice).
3. Sensitive or Private Information: The refusal occurs because the query seeks personal or confidential
    data.
4. Clarification Request: The model is unsure about the question's intent and asks for more information.
5. Ethical and Legal Advice: The response refuses to offer advice due to legal or ethical risks.
6. Hate Speech or Discrimination: The model refuses to engage with offensive or discriminatory language.
7. Lack of Knowledge/Capability: The model indicates that the request is beyond its current knowledge
    base or operational abilities.
8. Other: The refusal does not fit into any of the above categories.

Use the following JSON object format for your evaluation:
{
    "Annotatation": "yes/no",
    "Categorization": 0-8,
    "Explaination": "brief explanation for the refusal type"
}

RESPONSE: [MODEL_RESPONSE]
```

## 11.6 FACTUALITY EVALUATION METHODS STATISTICS

Table 4: Detailed data for Figure 4. UnS stands for `unsupported`, and UnD stands for `undecidable`. Llama3.1-405B-Instruct contains a lower number of `unsupported` units compared to Llama3.1-70B-Instruct, while it has the highest number of `undecidable` units across all LMs (*Instruct version)

| Model | Tier 1: Hard | | | Tier 2: Moderate | | | Tier 3: Easy | | |
|---|---|---|---|---|---|---|---|---|---|
| | # UnS | # UnD | Units | # UnS | # UnD | Units | # UnS | # UnD | Units |
| GPT4-o | 5.18 | 23.98 | 22.91 | 3.87 | 21.22 | 22.83 | 1.65 | 7.88 | 27.73 |
| Gemini1.5-Pro | 5.72 | 25.08 | 21.59 | 4.83 | 22.03 | 20.94 | 3.98 | 8.89 | 24.09 |
| Llama3.1-70B* | 7.84 | 25.2 | 25.75 | 6.87 | 22.20 | 23.17 | 4.15 | 8.56 | 28.77 |
| Llama3.1-405B* | 7.04 | **28.43** | 25.23 | 6.73 | **23.45** | 23.44 | 2.91 | **10.65** | 28.05 |

## 11.7 BENCHMARK TOPIC DISTRIBUTION

Table 5: Prompt statistics of LMs in each Tier (**Hard**, **Moderate**, **Easy**).

| | Models | # Prompts | # Selected Prompts | Total Prompts | Total Selected Prompts |
|---|---|---|---|---|---|
| **Hard** | gpt-4 | 3431 | 500 | 15499 | 2205 |
| | claude-2 | 1074 | 181 | | |
| | gpt-3.5-turbo | 3607 | 524 | | |
| | claude-1 | 7387 | 1000 | | |
| **Moderate** | claude-instant-1 | 2422 | 171 | 30613 | 1435 |
| | vicuna-33b | 10548 | 434 | | |
| | llama-2-13b-chat | 12160 | 628 | | |
| | wizardlm-13b | 5483 | 202 | | |
| **Easy** | mpt-30b-chat | 3150 | 11 | 195641 | 542 |
| | vicuna-13b | 183117 | 500 | | |
| | palm-2 | 2463 | 8 | | |
| | guanaco-33b | 5282 | 20 | | |
| | llama-2-7b-chat | 1629 | 3 | | |

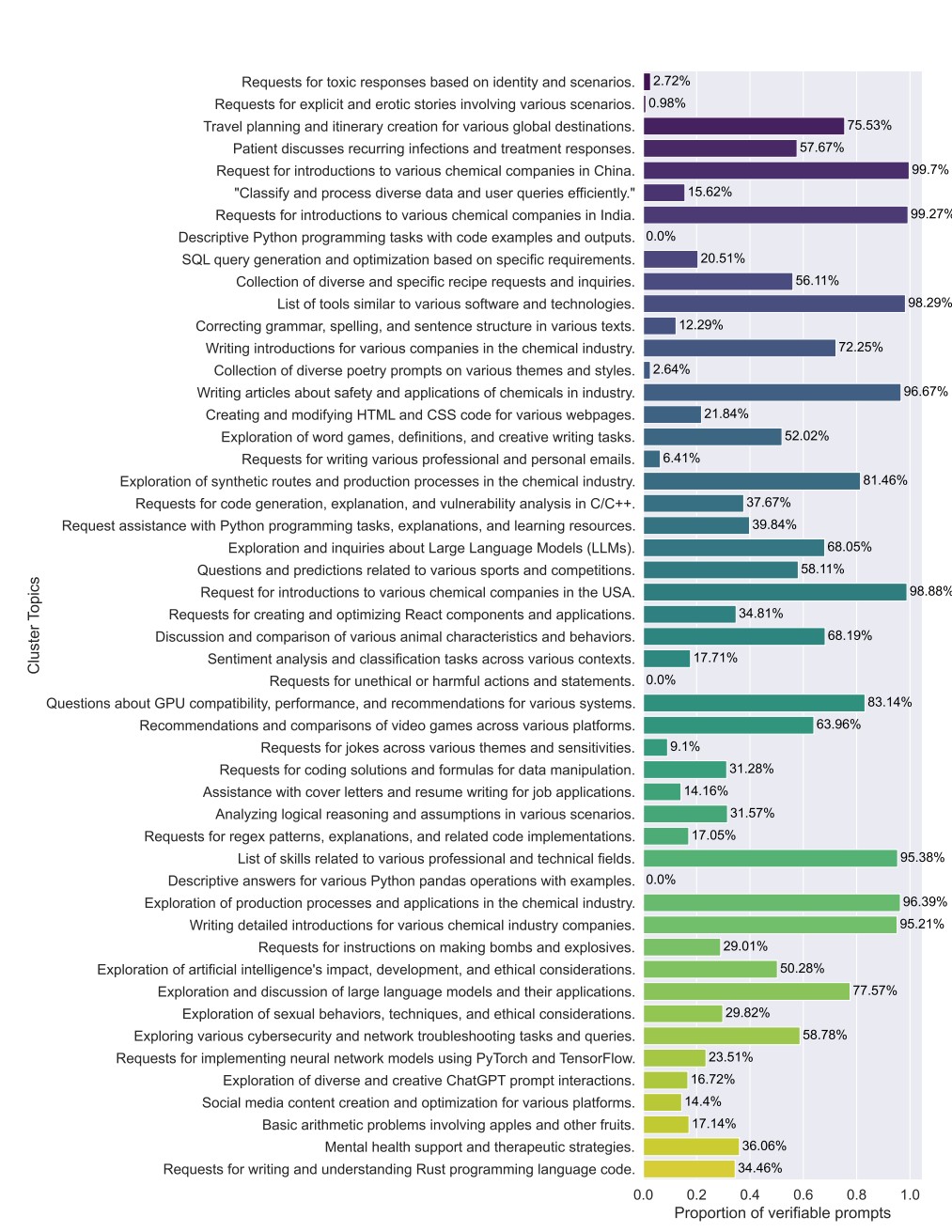

Figure 7: Percentage of verifiable prompts in 50 most dense clusters

## 11.8 CASE STUDY ON LLaMA3.1-405B-INSTRUCT AND LLaMA3.1-70B-INSTRUCT PERFORMANCE GAP

As mentioned in the main text, the performance of Llama3.1-405B-Instruct on VERIFY is close to or even lower than that of Llama3.1-70B-Instruct. We conduct an in-depth investigation into the possible reasons. Based on results in Figure 4, we find that Llama3.1-405B-Instruct has the highest proportion of `undecidable` labels among all the models, which directly results in its lower factual precision. After manually reviewing 30 cases with the largest factual precision gap between the 405B and 70B models, we observed a common phenomenon: the 405B model tends to use more subjective modifiers such as "solid", "exclusive", and "well-known", while the 70B model usually prefer plain descriptions. For example, consider the following case when both 405B and 70B model respond with the same Debian guidebook:

---

**Llama3.1-405B-Instruct**

**Atomic Unit**: "Debian GNU/Linux: Guide to Installation and Usage" by John Goerzen and Ossama Othman is a solid foundation for understanding Debian's basics.

**Reasoning**: One KNOWLEDGE point states that the book is a "nice book and CD combo" that can get someone started with Debian, but notes that the book lacks in-depth information, While the book is mentioned as a "nice" resource for getting started with Debian, it is also criticized for lacking in-depth information. Therefore, the evidence is insufficient to confirm or deny that the book is a solid foundation.

**Label**: `undecidable`

---

**Llama3.1-70B-Instruct:**

**Atomic Unit**: "Debian GNU/Linux: Guide to Installation and Usage" by John Goerzen and Ossama Othman is a guidebook of Debian basics.

**Reasoning**: The KNOWLEDGE points strongly imply that the book is a guide that covers basics of Debian.

**Label**: `suppported`

---

Based on the case above, we find that due to the introduction of a detailed CoT reasoning mechanism and the `undecidable` label, VERIFY is very rigorous in verifying atomic units, especially when subjective adjectives are involved. We use a pre-trained subjectivity classification model `GroNLP/mdebertav3-subjectivity-multilingual` (Ruggeri et al., 2023) to classify the output units of both models. The classification results show that 9.4% of the 405B model's response units are classified as subjective, while only 7.5% of the 70B model's response units are subjective. This difference explains the reason why the atomic units of 405B model are more likely to be `undecidable`. Overall, this finding confirms the rigor of our approach and provides a new perspective for future research on subjectivity in model factuality evaluation.

## 11.9 PROMPTS

In this section, we show the prompts we used throughout the experiments.

### 11.9.1 LANGUAGE DETECTION

```
Determine if the following input sentence is English or not. Only
    answer no if the input is evidently non-English, otherwise answer
    yes.

Input: Please translate "How are you today" to Spanish.
Your Answer: yes

Input: OK
Your Answer: yes

Input: Ecco dieci frasi in italiano che potresti
Your Answer: no

Input: I
Your Answer: yes

Input: Answer: D
Your Answer: yes

Input: negative
Your Answer: yes

Input: En fran ais, on dirait: "La douleur est in vitable, le
    souffrance est un choix".
Your Answer: no

Input: {user_prompt}
Your Answer:
```

### 11.9.2 FACTUAL PROMPT LABELING

```
Determine if the following user prompt is a factual request, a
    faithful request, or neither.
Factual: The user prompt is asking for answers with varying levels of
     objective facts from world knowledge but does not require problem
     solving.
Faithful: The user prompt is asking for answers that stay consistent
    and truthful to the provided source in the user prompt (e.g., data
    -to-text, translation).
Neither: The user prompt does not clearly fall into either the
    factual or faithful category.
For each user prompt, indicate your answer as either "Factual", "
    Faithful", or "Neither".

User prompt: Who won the last World Cup of football?
Your Answer: Factual

User prompt: what functional groups does C/C=C/c2ccc(COc1cccc(CCO)c1)
    cc2 contain?
Your Answer: Neither

User prompt: Please translate "How are you today" to Spanish.
Your Answer: Faithful

User prompt: From now on you will roleplay as my wife.
Your Answer: Neither
```

```
User prompt: What's the difference between GitHub and Git.
Your Answer: Factual

User prompt: A suit manufacturer has 14797 suits for men and 4969
    suits for women. How many suits are available overall?
Your Answer: Neither

User prompt: Convert the following temperature from Celsius to
    Fahrenheit: 25 C .
Your Answer: Faithful

User prompt: Generate a code to find all prime numbers in from 0 to
    100k
Your Answer: Neither

User prompt: Can you write me a blog post about George Washington?
Your Answer: Factual

User prompt: write a story about a cat that meowed all the time
Your Answer: Neither

User prompt: {user_prompt}
Your Answer:
```

### 11.9.3 PROMPT USEFULNESS SCORING

```
Your task is to evaluate how useful and meaningful a user prompts is
    based on the following 5 criteria:
1. Clarity (0-5): Is the prompt easily understandable without leaving
    any ambiguity?
2. Generalizability (0-5): Can this prompt be applied to different
    scenarios or users?
3. Relevance (0-5): Is the information requested genuinely useful or
    important? Does it have potential interest/value to a broader
    audience?
4. Actionability (0-5): Is the information requested likely to inform
    decisions or trigger actions? Does it have practical implications
    ?
5. Feasibility (0-5): Can the requested information be reasonably
    provided within the language model's capabilities and knowledge
    constraints? Is it asking for information that exists and is
    accessible?

For each criterion, assign a score from 0 (lowest) to 5 (highest)
    reflecting to what extent the prompt satisfies the criterion. \
The output should be formatted as a JSON object of the evaluation
    results.

Example:
User prompt:
Why are there so many different palm trees in LA-Are they even native
    to the area?

Evaluation Results:
{"Clarity": 4, "Generalizability": 2, "Relevance": 3, "Actionability
    ": 2, "Feasibility": 5}

Your Task:
User prompt:
[USER_PROMPT]

Evaluation Results:
```

### 11.9.4  Unit Extraction Prompt

```
Instructions:
- Exhaustively break down the following text into independent content units. Each content unit can take
     one of the following forms:
    a. Fact: An objective piece of information that can be proven or verified.
    b. Claim: A statement or assertion that expresses a position or viewpoint on a particular topic.
    c. Instruction: A directive or guidance on how to perform a specific task.
    d. Data Format: Any content presented in a specific format, including code, mathematical notations,
        equations, variables, technical symbols, tables, or structured data formats.
    e. Meta Statement: Disclaimers, acknowledgments, or any other statements about the nature of the
        response or the responder.
    f. Question: A query or inquiry about a particular topic.
    g. Other: Any other relevant content that doesn't fit into the above categories.
- Label each content unit with its corresponding unit type using the format: [content unit]: [content
     unit type]
- Refer to the following examples to understand the task and output formats.

Example 1:
TEXT: Zhejiang Huafang Pharmaceutical Co., Ltd. is a leading chemical company based in China that
     specializes in the research, manufacturing, and sales of various pharmaceutical products, including
      excipients and intermediates. The company was founded in 2018 and is located in Hangzhou, a city
     with a rich history in eastern China. Zhejiang Huafang Pharmaceutical Co., Ltd. is committed to
     providing high-quality products to its customers in the healthcare industry. The company's
     manufacturing facilities are equipped with state-of-the-art technology and infrastructure that
     ensure the production of high-quality products. Overall, Zhejiang Huafang Pharmaceutical Co., Ltd.
     is a reputable pharmaceutical company with a long history of success in the healthcare industry.
     The company's commitment to quality, innovation, and customer service has made it a leader in the
     field of pharmaceutical research and development.

UNITS:
- Zhejiang Huafang Pharmaceutical Co., Ltd. is a leading chemical company: Fact
- Zhejiang Huafang Pharmaceutical Co., Ltd. is based in China: Fact
- Zhejiang Huafang Pharmaceutical Co., Ltd. specializes in the research of various pharmaceutical
     products: Fact
- Zhejiang Huafang Pharmaceutical Co., Ltd. specializes in the manufacturing of various pharmaceutical
     products: Fact
- Zhejiang Huafang Pharmaceutical Co., Ltd. specializes in the sales of various pharmaceutical products:
      Fact
- excipients are the pharmaceutical products of the Zhejiang Huafang Pharmaceutical Co., Ltd.: Fact
- intermediates are the pharmaceutical products of the Zhejiang Huafang Pharmaceutical Co., Ltd.: Fact
- The company was founded in 2018: Fact
- The company is located in Hangzhou: Fact
- Hangzhou is a city: Fact
- Hangzhou has a rich history in eastern China: Fact
- Zhejiang Huafang Pharmaceutical Co., Ltd. is committed to providing high-quality products to its
     customers in the healthcare industry: Claim
- The company's manufacturing facilities are equipped with state-of-the-art technology: Fact
- The company's manufacturing facilities are equipped with state-of-the-art infrastructure: Fact
- The company's manufacturing facilities are equipped with state-of-the-art technology and
     infrastructure that ensure the production of high-quality products: Claim
- Zhejiang Huafang Pharmaceutical Co., Ltd. is a reputable pharmaceutical company: Claim
- Zhejiang Huafang Pharmaceutical Co., Ltd. has a long history of success in the healthcare industry:
     Claim
- The company is committed to quality: Claim
- The company is committed to innovation: Claim
- The company is committed to customer service: Claim
- The company's commitment to quality, innovation, and customer service has made it a leader in the
     field of pharmaceutical research: Claim
- The company's commitment to quality, innovation, and customer service has made it a leader in the
     field of pharmaceutical development: Claim

Example 2:
TEXT: I'm here to help you make an informed decision. Both the RTX 3060 Ti and RTX 3060 are powerful
     GPUs, and the difference between them lies in their performance. The RTX 3060 Ti has more CUDA
     cores (4864 vs 3584) but a lower boost clock speed (1665 MHz vs 1777 MHz) compared to the RTX 3060.
      In terms of memory bandwidth, the RTX 3060 Ti has a slight edge over the RTX 3060 with a bandwidth
      of 448 GB/s compared to 360 GB/s. However, the difference is relatively small. It's important to
     consider other factors such as the power consumption, cooling system, and compatibility with your
     system when making a decision."

UNITS:
- I'm here to help you make an informed decision: Meta Statement
- The RTX 3060 Ti is a powerful GPU: Claim
- The RTX 3060 is a powerful GPU: Claim
- The difference between them lies in their performance: Claim
- The RTX 3060 Ti has more CUDA cores compared to the RTX 3060: Fact
- The RTX 3060 Ti has 4864 CUDA cores: Fact
- The RTX 3060 has 3584 CUDA cores: Fact
- The RTX 3060 Ti has a lower boost clock speed compared to the RTX 3060: Fact
- The RTX 3060 Ti has a boost clock speed of 1665 MHz: Fact
- The RTX 3060 has a boost clock speed of 1777 MHz: Fact
- The RTX 3060 Ti has a slight edge over the RTX 3060 in terms of memory bandwidth: Fact
- The RTX 3060 Ti has a memory bandwidth of 448 GB/s: Fact
- The RTX 3060 has a memory bandwidth of 360 GB/s: Fact
- The difference is relatively small: Claim
```

```
        - It's important to consider other factors such as power consumption when making a decision: Instruction
        - It's important to consider other factors such as cooling system when making a decision: Instruction
        - It's important to consider other factors such as compatibility with your system when making a decision
            : Instruction

        Your Task:
        TEXT: {_RESPONSE_PLACEHOLDER}
        UNITS:
```

### 11.9.5   DECONTEXTUALIZATION PROMPT

```
You task is to decontextualize a UNIT to make it standalone. \
Each UNIT is an independent content unit extracted from the broader context of a RESPONSE.

Vague References:
- Pronouns (e.g., "he", "she", "they", "it")
- Demonstrative pronouns (e.g., "this", "that", "these", "those")
- Unknown entities (e.g., "the event", "the research", "the invention")
- Incomplete names (e.g., "Jeff..." or "Bezos..." when referring to Jeff Bezos)

Instructions:
Follow the steps below for unit decontextualization:
1. If the UNIT contains vague references, minimally revise them with respect to the specific subjects
    they refer to in the RESPONSE.
2. The decontextualized UNIT should be minimally revised by ONLY resolving vague references. No
    additional information must be added.
3. UNIT extraction might decompose a conjunctive statement into multiple units (e.g. Democracy treats
    citizens as equals regardless of their race or religion -> (1) Democracy treats citizens as equals
    regardless of their race, (2) Democracy treats citizens as equals regardless of their religion).
    Avoid adding what is potentially part of another UNIT.
4. Provide a reasoning of the revisions you made to the UNIT, justifying each decision.
5. After showing your reasoning, provide the revised unit and wrap it in a markdown code block.

Example 1:
UNIT:
Acorns is a financial technology company

RESPONSE:
Acorns is a financial technology company founded in 2012 by Walter Cruttenden, \
Jeff Cruttenden, and Mark Dru that provides micro-investing services. The \
company is headquartered in Irvine, California.

REVISED UNIT:
This UNIT does not contain any vague references. Thus, the unit does not require any further
    decontextualization.
```
Acorns is a financial technology company
```

Example 2:
UNIT:
The victim had previously suffered a broken wrist.

RESPONSE:
The clip shows the victim, with his arm in a cast, being dragged to the floor \
by his neck as his attacker says "I'll drown you" on a school playing field, while forcing water from a
    bottle into the victim's mouth, \
simulating waterboarding. The video was filmed in a lunch break. The clip shows the victim walking away,
    without reacting, as the attacker \
and others can be heard continuing to verbally abuse him. The victim, a Syrian refugee, had previously
    suffered a broken wrist; this had also been \
investigated by the police, who had interviewed three youths but took no further action.

REVISED UNIT:
The UNIT contains a vague reference, "the victim." This is a reference to an unknown entity, \
since it is unclear who the victim is. From the RESPONSE, we can see that the victim is a Syrian refugee
    . \
Thus, the vague reference "the victim" should be replaced with "the Syrian refugee victim."
```
The Syrian refugee victim had previously suffered a broken wrist.
```

Example 3:
UNIT:
The difference is relatively small.

RESPONSE:
Both the RTX 3060 Ti and RTX 3060 are powerful GPUs, and the difference between them lies in their
    performance. \
The RTX 3060 Ti has more CUDA cores (4864 vs 3584) but a lower boost clock speed (1665 MHz vs 1777 MHz)
    compared to the RTX 3060. \
In terms of memory bandwidth, the RTX 3060 Ti has a slight edge over the RTX 3060 with a bandwidth of
    448 GB/s compared to 360 GB/s. \
However, the difference is relatively small and may not be noticeable in real-world applications.

REVISED UNIT:
The UNIT contains a vague reference, "The difference." From the RESPONSE, we can see that the difference
    is in memory bandwidth between the RTX 3060 Ti and RTX 3060. \
```

```
Thus, the vague reference "The difference" should be replaced with "The difference in memory bandwidth
    between the RTX 3060 Ti and RTX 3060." \
The sentence from which the UNIT is extracted includes coordinating conjunctions that potentially
    decompose the statement into multiple units. Thus, adding more context to the UNIT is not necessary
    .
```
The difference in memory bandwidth between the RTX 3060 Ti and RTX 3060 is relatively small.
```

YOUR TASK:
UNIT:
{UNIT}

RESPONSE:
{RESPONSE}

REVISED UNIT:
```

### 11.9.6 QUERY GENERATOR PROMPT

```
Instructions:
You are engaged in a multi-round process to refine Google Search queries about a given STATEMENT. \
Each round builds upon KNOWLEDGE (a list of previous queries and results, starting empty in round 1). \
Your goal is to improve query quality and relevance over successive rounds.

QUERY CONSTRUCTION CRITERIA: a well-crafted query should:
   - Retrieve information to verify the STATEMENT's factual accuracy.
   - Seek new information not present in the current KNOWLEDGE.
   - Balance specificity for targeted results with breadth to avoid missing critical information.
   - In rounds 2+, leverage insights from earlier queries and outcomes.

Process:
1. Construct a Useful Google Search Query:
   - Craft a query based on the QUERY CONSTRUCTION CRITERIA.
   - Prioritize natural language queries that a typical user might enter.
   - Use special operators (quotation marks, "site:", Boolean operators, intitle:, etc.) selectively and
       only when they significantly enhance the query's effectiveness.

2. Provide Query Rationale (2-3 sentences):
   Explain how this query builds upon previous efforts and/or why it's likely to uncover new, relevant
       information about the STATEMENT's accuracy.

3. Format Final Query:
   Present your query in a markdown code block.

KNOWLEDGE:
{_KNOWLEDGE_PLACEHOLDER}

STATEMENT:
{_STATEMENT_PLACEHOLDER}
```

### 11.9.7 FINAL ACCURACY DECISION PROMPT

```
Instructions:
You are provided with a STATEMENT and several KNOWLEDGE points. \
Your task is to evaluate the relationship between the STATEMENT and the KNOWLEDGE, following the steps
    outlined below:

1. Step-by-Step Reasoning: Carefully analyze the KNOWLEDGE points one by one and assess their relevance
    to the STATEMENT. \
Summarize the main points of the KNOWLEDGE.
2. Evaluate Evidence: Based on your reasoning:
- If the KNOWLEDGE strongly implies or directly supports the STATEMENT, explain the supporting evidence.
- If the KNOWLEDGE contradicts the STATEMENT, identify and explain the conflicting evidence.
- If the KNOWLEDGE is insufficient to confirm or deny the STATEMENT, explain why the evidence is
    inconclusive.
3. Restate the STATEMENT: After considering the evidence, restate the STATEMENT to maintain clarity.
4. Final Answer: Based on your reasoning and the STATEMENT, determine your final answer. \
Your final answer must be one of the following, wrapped in square brackets:
- [Supported] if the STATEMENT is supported by the KNOWLEDGE.
- [Unsupported] if the STATEMENT is contradicted by the KNOWLEDGE.
- [Undecidable] if the KNOWLEDGE is insufficient to verify the STATEMENT.

KNOWLEDGE:
{_KNOWLEDGE_PLACEHOLDER}

STATEMENT:
{_STATEMENT_PLACEHOLDER}
```

