# OpenReview forum: "FactBench: A Dynamic Benchmark for In-the-Wild Language Model Factuality Evaluation"
_ICLR.cc/2025/Conference — ICLR 2025 Conference Withdrawn Submission_

### Official Review · Reviewer_42L8 · 2024-11-01

**Soundness:** 2
**Presentation:** 2
**Contribution:** 3
**Rating:** 5
**Confidence:** 4

**Summary:**

This paper introduces VERIFY, an automatic factuality evaluation pipeline that assesses language model (LM) responses by verifying and categorizing content units based on web evidence. Using VERIFY, the authors built FACTBENCH, a benchmark of 985 prompts across various topics and difficulty levels to evaluate LMs' accuracy in real-world settings. Results show that VERIFY aligns closely with human judgment, establishing it as a reliable method for assessing factuality in LM outputs.

**Strengths:**

The task of factuality evaluation is extremely important in large language models (LLMs), as ensuring the factual accuracy of model outputs is crucial for their reliability in real-world applications. The proposed approach of categorizing LM-generated content into "supported," "unsupported," or "undecidable" represents a novel and rigorous method, distinguishing this work from previous studies that often lack this level of granularity. This categorization allows for a more nuanced understanding of model limitations and strengths in factual reasoning. Furthermore, the authors carefully selected datasets and established comprehensive baselines, ensuring that their comparisons across methods are both fair and robust.

**Weaknesses:**

(1) Building FACTBENCH requires multiple steps, yet each step relies on relatively simple, heuristic approaches, which may limit the novelty and robustness of the benchmark in capturing nuanced factuality challenges.

(2) The lack of qualitative analysis for ambiguous samples or Tier 1 (Hard) prompts leaves gaps in interpreting how VERIFY handles especially difficult cases. Providing a more detailed examination of these prompts and their handling would enrich the analysis.

(3) While VERIFY’s methodology is strong in categorizing factuality, it is relatively limited in interpretability when applied to complex or context-dependent responses. Expanding on how VERIFY’s outputs could be used to provide actionable feedback for model improvement or on how it handles responses with interdependent factual claims would increase its utility and depth.

**Questions:**

(1) There is a typo on line 305: it references Table 2 when it should be Figure 2.

(2) The high refusal rate observed in Gemini1.5-Pro, especially in the Hard tier, is mentioned briefly in the paper. Could the authors provide more details for these refusals and any insights into how different refusal categories might impact the overall factuality evaluation? Also, in cases where refusals were misclassified or unjustified, did the authors investigate how these instances were distributed across prompts or tiers?

(3) Did you verify whether calculating the score using the Hallucination Score as defined in Eq. 1 is indeed meaningful?

---

> ### Author Response · Authors · 2024-11-20
> **Response to Reviewer 42L8 (Part 1/2)**
>
> Thank you for your thorough and insightful review of our paper. We appreciate your detailed feedback and your constructive suggestions. Below, we address your concerns:
>
> **Weaknesses:**
> > (1) Building FACTBENCH requires multiple steps, yet each step relies on relatively simple, heuristic approaches, which may limit the novelty and robustness of the benchmark in capturing nuanced factuality challenges.
>
> The task of evaluating long-form text is challenging, and there have been many efforts to formulate it. We follow the previous work [1, 2] to break this task into a pipeline of steps, each designed with a specific goal and working towards the purpose of factual verification. Our evaluation method is the first that can identify the verifiable content during user-model interactions, check their factuality through multiple rounds of evidence-retrieval, and use the hallucination degree in the LM responses as a way to measure the likelihood of prompt-eliciting hallucinations from LMs.
> We would also appreciate it if you could elaborate more on **nuanced factuality challenges**.
>
> >(2) The lack of qualitative analysis for ambiguous samples or Tier 1 (Hard) prompts leaves gaps in interpreting how VERIFY handles especially difficult cases. A more detailed examination of these prompts and their handling would enrich the analysis.
>
> We acknowledge the need for a thorough qualitative analysis of challenging cases and will include additional examples of our qualitative analysis. Meanwhile, here are illustrative examples that demonstrate VERIFY's handling of difficult judgments, comparing its evaluation results with FactCheck-GPT on responses generated by Llama3.1-405B-Instruct. We focus on comparing with FactCheck-GPT since other baselines showed significantly weaker performance.
>
>
> **prompt:** "What are the best solutions for quickly and easily testing prompts with multiple different LLM models/services?
>
> **response:** "Testing prompts with multiple LLM (Large Language Model) models/services can be a time-consuming task, but there are several solutions that can make it quicker and easier. Here are some of the best solutions:
> 1. **LLaMA**: LLaMA (Large Language Model Application) is an open-source framework that allows you to test prompts with multiple LLM models, including those from Hugging Face, Google, and Microsoft. LLaMA provides a simple API for testing prompts and supports multiple input formats.
> 2. **Prompt Engineering**: Prompt Engineering is a platform that allows you to design, test, and optimize prompts for multiple LLM models. It provides a visual interface for crafting prompts and supports testing with models from Hugging Face, Google, and Microsoft.
>
>
> **Unit 1:** LLaMA supports models from Hugging Face, Google, and Microsoft.
>
> FactCheck-GPT: Supported
>
> VERIFY: Unsupported
>
> ***Analysis***: VERIFY correctly identified this technical inaccuracy, as LLaMA is a standalone model family that does not integrate with these platforms.
>
> **Unit 2:** LLaMA supports multiple input formats
>
> FactCheck-GPT: Supported
>
> VERIFY: Undecidable
>
> ***Analysis:*** VERIFY appropriately flagged this as undecidable since input format support varies across LLaMA versions and implementations.
>
> **Unit 3:** Prompt Engineering is a platform that allows you to design, test, and optimize prompts for multiple LLM models.
>
> FactCheck-GPT: Supported
>
> VERIFY: Contradicted
>
> ***Analysis:*** VERIFY correctly identified that prompt engineering is a methodology, not a platform, showing its ability to distinguish conceptual differences.
>
> **Unit 4:** Prompt Engineering supports testing with models from Hugging Face, Google, and Microsoft.
>
> FactCheck-GPT: Supported
>
> VERIFY: Undecidable
>
> ***Analysis:*** VERIFY correctly labeled this as undecidable since prompt engineering, as a methodology, can be applied to any model without having explicit "support."
>
> These examples showcase VERIFY's ability to handle technical inaccuracies, ambiguities, and conceptual distinctions in factual verification.
>
> > (3) While VERIFY’s methodology is strong in categorizing factuality, it is relatively limited in interpretability when applied to complex or context-dependent responses. Expanding on how VERIFY’s outputs could be used to provide actionable feedback for model improvement or on how it handles responses with interdependent factual claims would increase its utility and depth.
>
> Thank you for the insightful suggestion. Although out of the scope of this work, these are promising future directions that we plan to pursue in the follow-up work.

---

> ### Author Response · Authors · 2024-11-20
> **Response to Reviewer 42L8 (Part 2/2)**
>
> **Questions:**
> > (1) There is a typo on line 305: it references Table 2 when it should be Figure 2.
>
> Thanks for pointing this out, we will modify this in the final version.
>
> > (2) The high refusal rate observed in Gemini1.5-Pro, especially in the Hard tier, is mentioned briefly in the paper. Could the authors provide more details for these refusals and any insights into how different refusal categories might impact the overall factuality evaluation? Also, in cases where refusals were misclassified or unjustified, did the authors investigate how these instances were distributed across prompts or tiers?
>
> Thank you for your insightful question regarding the refusal rates observed in Gemini1.5-Pro and their impact on our factuality evaluation. We appreciate the opportunity to provide more details and context on this important aspect of our study.
> We conducted a detailed analysis of these refusals, categorizing them into different types according to the LM response (appendix 12.6 details the breakdown of refusal categories). Here's a breakdown of our findings across tiers and refusal categories:
>
> **Hard Tier:**
>
> Highest number of refusals: 28 total
>
> Dominated by misinformation risks (16 instances)
>
> Followed by ethical/legal advice concerns (6 instances) and safety concerns (6 instances)
>
> **Moderate Tier:**
>
> Fewer refusals compared to the Hard tier: 12 total
>
> 5 instances for ethical/legal advice, 4 for misinformation risks, and 3 for safety concerns
>
> **Easy Tier:**
>
> Lowest refusal rate: Only 1 instance
>
> Single refusal related to ethical/legal advice
>
> This distribution shows a clear correlation between query difficulty and refusal frequency, with the Hard tier prompting the most cautious responses from Gemini1.5-Pro.
> |Tier|Misinformation Risks|Ethical and Legal Advice|Safety Concerns|Total Refusals|
> |--|--|--|--|--|
> |Hard|16|6|6|28|
> |Moderate|4|5|3|12|
> |Easy|0|1|0|1|
>
> **Misclassification Analysis:**
>
> |Tier| Category| Refusals|Misclassified|
> |----|--------|--------|-------------|
> |Hard|Misinformation Risks|16|10|
> |Hard|Ethical and Legal Advice|6|6|
> |Hard|Safety Concerns|6|2|
> |Moderate|Misinformation Risks|4|2|
> |Moderate|Ethical and Legal Advice|5|2|
> |Moderate|Safety Concerns|3|2|
> |Easy|Ethical and Legal Advice|1|1|
>
> The data shows that Label 5 (Ethical and Legal Advice) was most prone to misclassification across all tiers. This suggests that Gemini1.5-Pro tends to overestimate ethical and legal risks in queries, leading to unnecessary refusals. Label 2 (Misinformation Risks) also showed a high misclassification rate, particularly in the Hard tier, indicating that the model may be overly cautious in assessing potential misinformation, especially for complex queries.
>
> > (3) Did you verify whether calculating the score using the Hallucination Score as defined in Eq. 1 is indeed meaningful?
>
> Our score incorporates both unsupported and undecidable units to address the complexities of evaluating model generations. Unsupported units represent clear inaccuracies, as they are verifiably incorrect based on evidence. Undecidable units, on the other hand, may be either correct or incorrect due to several factors: (1) knowledge from the model's training data that could be accurate but lacks sufficient context or might be inaccurate if outdated or misconstrued; (2) information that is not readily verifiable through current web searches, which might either be factual or fabricated; and (3) novel combinations of facts that appear plausible but are either incorrect or undocumented.
>
> The weighting factor α balances the importance of undecidable and unsupported units. To determine the appropriate α value, we analyzed 100 responses (25 per model). Two annotators evaluated 570 undecidable units, achieving strong inter-annotator agreement (85.5%). Across all models, 57% of undecidable units were found to be factual and 43% not factual, with individual models showing similar patterns as shown in the table below. Based on this finding, we set α = 0.5.
> |Model|Factual(Avg.Percentage)|NotFactual(Avg.Percentage)|
> |-----|-----------------------|--------------------------|
> |GPT4-o|68.4|31.6|
> |Gemini1.5-Pro|56.6|43.4|
> |Llama3.1-405B-Instruct|51.0|49.0|
> |Llama3.1-70B-Instruct|52.0|48.0|
> |**Average**|**57.0**|**43.0**|
>
> **References**
> 1. Min, S., et al. 2023. FActScore: Fine-grained atomic evaluation of factual precision in long-form text generation. In Proceedings of the 2023 Conference on Empirical Methods in Natural Language Processing.
> 2. Wei, J., et al. 2024. Long-form factuality in large language models. In 2024 Conference on Neural Information Processing Systems.
> 3. Wang, Y., et al. 2024. Factcheck-bench: Fine-grained evaluation benchmark for automatic fact-checkers. In Findings of the Association for Computational Linguistics: EMNLP 2024.

---

> > ### Comment · Reviewer_42L8 · 2024-11-25
> >
> > Thank you for your detailed response, which clarified the methodology and design of VERIFY. While the approach shows potential, I still have two main concerns:
> >
> > 1. Reliance on Heuristic Methods: VERIFY’s dependence on heuristic-based processes may limit its robustness, particularly when addressing nuanced or complex scenarios such as ambiguous prompts or Hard tier cases.
> >
> > 2. Fixed α Value: Although the choice of an α value of 0.5 is supported by analysis, using a fixed value risks oversimplifying the variability in factuality across different models and prompts. A dynamic weighting approach might better account for the diverse distributions of undecidable units.

---

> > > ### Author Response · Authors · 2024-11-25
> > > **Response to Reviewer 42L8**
> > >
> > > Thank you for your response. We have provided further clarifications regarding your concerns.
> > > > Reliance on Heuristic Methods: VERIFY’s dependence on heuristic-based processes may limit its robustness, particularly when addressing nuanced or complex scenarios such as ambiguous prompts or Hard tier cases.
> > > 1. Our usefulness score evaluates whether the prompt is clear and understandable; therefore, ambiguous prompts have not been the focus of this work. Also, please check the prompt we investigated for qualitative analysis which is selected from the *Hard* tier.
> > > 2. VERIFY is the first method that can effectively and efficiently extract verifiable information from daily user-model conversations, as supported by evaluation results and qualitative analysis. Long-form factuality evaluation is a complex task that must be broken down into simple steps. Asking a language model to evaluate the response at once leads to inaccurate and coarse-level evaluations that lack interpretability.
> > >
> > > > Fixed α Value: Although the choice of an α value of 0.5 is supported by analysis, using a fixed value risks oversimplifying the variability in factuality across different models and prompts. A dynamic weighting approach might better account for the diverse distributions of undecidable units.
> > >
> > > Thanks for your suggestion. While we acknowledge that different models and prompt types may require their own optimal α values, α selection heavily relies on the model's internal knowledge. For any new model or prompt type, determining the optimal α requires human annotation, which we believe contradicts the original purpose of automating the factuality evaluation. Results of per-model annotation show similar α values, suggesting a value of 0.5 is a reasonable estimation across models and balancing cost-effectiveness and performance.

---

### Official Review · Reviewer_BPdM · 2024-11-01

**Soundness:** 3
**Presentation:** 3
**Contribution:** 3
**Rating:** 5
**Confidence:** 3

**Summary:**

This paper introduces FactBench, a dynamic benchmark designed to evaluate the factual accuracy of large language models (LMs) in real-world contexts. FactBench continuously updates its dataset of prompts to capture scenarios where LMs are likely to generate hallucinations, addressing the limitations of existing static benchmarks. Additionally, the VERIFY framework is presented as a factuality assessment pipeline that categorizes LM responses based on evidence-supported verification categories, demonstrating higher alignment with human evaluations.

**Strengths:**

● The paper proposes a benchmark across multiple topics with varying difficulty levels.
● It offers an interesting categorization of hallucinations, distinguishing between context-independent and context-dependent statements, which could facilitate finer-grained hallucination detection in future studies.
● A new weighting factor is introduced to account for unsupported and undecidable units, adding robustness to hallucination scoring.
● The VERIFY framework shows good alignment with human judgment, particularly in handling nuanced cases.

**Weaknesses:**

● The experimental design could be improved; using VERIFY to classify data and then evaluate it may introduce circularity in the results.
● The VERIFY method lacks innovation in hallucination detection, particularly in terms of recall, which is essential in high-stakes fields like medical and encyclopedic contexts where hallucinations must be minimized.
● Difficulty ratings for prompts based solely on scores from multiple large models are unconvincing; a broader and more comprehensive classification method is needed.

**Questions:**

● Could you explain in detail how Llama3-70B was used to determine whether the data was verifiable?
● Why did you choose 0.5 as the weighting factor for the hallucination score? Might this choice impact the correlation of VERIFY with human preferences?

Comments:

Overall, this paper makes meaningful contributions to the development of factuality evaluation methods for LMs, particularly in establishing a dynamic benchmark that could adapt to future model advancements. However, improvements in experimental design, verification transparency, and difficulty categorization would enhance the robustness and generalizability of the findings.

---

> ### Author Response · Authors · 2024-11-20
> **Response to Reviewer BPdM**
>
> Thank you for your thorough and insightful review of our paper. We appreciate your detailed feedback and your constructive suggestions. Below, we address your concerns:
>
> **Weaknesses**
> > ●  The experimental design could be improved; using VERIFY to classify data and then evaluate it may introduce circularity in the results.
>
> We appreciate the opportunity to clarify our methodology and address concerns about potential circularity in our design. Here is how VERIFY is used and validated in our study:
>
> 1. **Prompt Classification:** Prompts are classified into tiers based on their corresponding LM’s performance ranking from the Arena Leaderboard [1], which is completely separate from VERIFY's factuality assessments.
> 2. **Validation Process:** VERIFY's factuality judgments are independently validated against human annotations, with results showing a strong correlation and thus reinforcing the reliability of VERIFY’s judgments.
>
> By separating the tier classification process from the evaluation and incorporating human validation, we address any concerns around circularity.
>
> > ●  The VERIFY method lacks innovation in hallucination detection, particularly in terms of recall, which is essential in high-stakes fields like medical and encyclopedic contexts where hallucinations must be minimized.
>
> In this work, we design a factuality labeling framework that annotates verifiable units from real-world user-model conversations. We recognize that recall measurement remains challenging with open-ended questions in our FACTBENCH dataset. For example, if the user is asking for movie recommendations, it's difficult to determine the set of factual statements that could be made about each movie. A model might mention accurate details about a film while omitting others, making traditional recall metrics hard to apply. While our method shows a strong correlation with human judgments on the statements it does evaluate, we acknowledge this limitation, especially for high-stakes applications, and intend to explore this in future works.
>
> > ●  Difficulty ratings for prompts based solely on scores from multiple large models are unconvincing; a broader and more comprehensive classification method is needed.
>
> While our model scoring-based proxy has limitations, the empirical results from 7 popular models validate its utility as prompts classified as "hard" consistently elicit more non-factual outputs than medium and easy prompts. However, we recognize that a more robust classification system could incorporate additional dimensions like prompt complexity, model capability demands, domain expertise requirements, and task types. Such a multi-faceted approach would provide a more comprehensive and reliable measure of prompt difficulty beyond performance scores alone.
>
> **Questions:**
> > ● Could you explain in detail how Llama3-70B was used to determine whether the data was verifiable?
>
> We want to clarify that determining whether the prompt is verifiable serves as a filtering step to reduce the computational burden of the subsequent pipeline. In this step, we identify user prompts that request answers with varying levels of objective facts derived from world knowledge. For example, unverifiable prompts such as “Role play as my wife,” “Write a story about a cat that meowed all the time,” or “Generate a code to find all prime numbers” are filtered out at this stage. We include a detailed definition of a “verifiable” prompt (we also call them “factual” prompt) and a series of examples to guide Llama3-70B in completing this classification task. The specific prompt used can be found in the appendix 12.10.2.
>
> > ● Why did you choose 0.5 as the weighting factor for the hallucination score? Might this choice impact the correlation of VERIFY with human preferences?
>
> Thanks for raising this question. The weighting factor α balances the importance of undecidable and unsupported units. It accounts for LMs generating content from training data, which may include factual information not found in current Web searches. To choose the α value, we sampled 100 responses (25 per model) and asked 2 annotators to annotate whether units labeled as “undecidable” by VERIFY are supported against external evidence. Each annotator evaluates 570 units, and the inter-annotation agreement is 85.5%. The following table summarizes the portion of undecidable units that were labeled as factual vs not factual. As seen in this table, the undecidable units have a near-equal chance of being categorized as factual or not factual. Based on this finding, we set the α to 0.5 for the experiments.
>
> |Model|Factual (Avg. Percentage)|Not Factual (Avg. Percentage)|
> |-----|-------------------------|----------------------------|
> |GPT4-o|68.4|31.6|
> |Gemini 1.5-Pro|56.6|43.4|
> |Llama 3.1-405B-Instruct|51.0|49.0|
> |Llama 3.1-70B-Instruct|52.0|48.0|
> |**Average**|**57.0**|**43.0**|
>
> **Reference**
> 1. Arena Leaderboard. https://huggingface.co/spaces/lmsys/chatbot-arena-leaderboard

---

### Official Review · Reviewer_mXTb · 2024-11-03

**Soundness:** 2
**Presentation:** 3
**Contribution:** 3
**Rating:** 5
**Confidence:** 3

**Summary:**

The authors propose FACTBENCH, a dynamic benchmark for evaluating language model factuality in real-world scenarios by using prompts that often provoke hallucinations. They also present VERIFY, a pipeline that assesses factuality by categorizing responses as supported, unsupported, or undecidable, based on retrieved web evidence. The experiments show that VERIFY can achieve the highest correlation with human judgments.

**Strengths:**

1.	This study addresses a growing concern in the factuality of LM-generated content, especially in the context of hallucinations.
2.	FACTBENCH is an innovative, dynamic benchmark that adapts to new factuality challenges. And VERIFY is an innovative factuality evaluation approach that achieves more precise, human-aligned assessments.
3.	Extensive experiments across multiple models show that VERIFY aligns closely with human judgments.

**Weaknesses:**

1.	Only three speakers were hired for annotation, with relatively low inter-annotator agreement (Cohen’s Kappa scores of 0.52 and 0.55). The difference in factuality labeling between VERIFY and Factcheck-GPT is minor in Table 2 (≤ 0.01 for Factual labels), and VERIFY's performance is noticeably lower in Tables 1 and 3, suggesting potential limitations in annotation reliability and robustness.
2.	While FACTBENCH is described as dynamic, the paper lacks specifics on its update process, such as the conditions under which prompts are added or removed, the frequency of updates, and criteria for integrating new prompts.
3.	The paper provides limited discussion on key hyperparameters, such as α used in the hallucination score and the number of evidence retrieval rounds in VERIFY.

**Questions:**

1.	What factors influenced the decision to categorize prompts into three tiers?
2.	Why was a topic model-based approach (BERTopic) chosen over a general clustering method? Given that BERTopic parameters can affect clustering quality, how were these parameters tuned, and might they influence the final benchmark results?
3.  See the weakness.

---

> ### Author Response · Authors · 2024-11-20
> **Response to Reviewer mXTb (Part 1/2)**
>
> Thank you for your thorough and insightful review of our paper. We appreciate your detailed feedback and your constructive suggestions. Below, we address your concerns:
>
> **Weaknesses:**
> > **1.1**  Only three speakers were hired for annotation, with relatively low inter-annotator agreement (Cohen’s Kappa scores of 0.52 and 0.55)
>
> While it's true that we employed three annotators, it's important to contextualize Cohen's Kappa as a conservative measure that adjusts for chance agreement and tends to penalize discrepancies more heavily than raw agreement percentages. This can result in seemingly lower Kappa scores despite higher raw agreement.
>
> Specifically:
> - For independence labeling, annotators achieved an **80.9%** raw agreement, corresponding to a Kappa score of 0.52. Independence labeling is inherently subjective as it involves interpretive human judgment, making a Kappa score of 0.52 significant in this context.
> - For factuality labeling, the raw agreement was even higher at **84.8%**, with a corresponding Kappa score of 0.55.
>
> These raw agreement percentages reflect strong consistency among annotators and serve as a useful reference, even if Kappa scores provide a more rigorous assessment.
>
> > **1.2**  The difference in factuality labeling between VERIFY and Factcheck-GPT is minor in Table 2 (≤ 0.01 for Factual labels)
>
> We argue that the lower difference in factuality is attributed to the fact that Factcheck-GPT relies on its backbone LM’s parametric knowledge to verify some units if no evidence is retrieved. However, this reliance on parametric knowledge, though leading to similar factual precision scores, presents key limitations. Parametric knowledge can become outdated and may not generalize well to new information, making it less reliable than VERIFY's search-based approach, which consistently grounds verification in current external evidence. For instance, when Factcheck-GPT fails to retrieve evidence for the statement *"AI21 Labs' platform provides an interface for crafting prompts"*, it resorts to its parametric knowledge and labels this statement as not factual, whereas VERIFY can validate the claim by successfully retrieving and analyzing recent web articles.
>
> > **1.3** “VERIFY's performance is noticeably lower in Tables 1 and 3, suggesting potential limitations in annotation reliability and robustness.”
>
> We want to clarify that these factuality precisions are **not comparable** across different factuality evaluation methods for two reasons:
> 1. The factual precisions are not related to the strength of the evaluation method. For example, VERIFY’s factuality precision of 71.58% only indicates that VERIFY labels 71.58% of units in LM responses as factual. No ground-truth labels are involved in this process.
> 2. Each evaluation method has its unit extractor, and therefore, the results in Table 1 highly depend on the granularity and quality of the units extracted. This motivated us to conduct the human evaluation experiments in Section 7.2, where all factuality evaluation methods are fed the same content units to verify.
>
> Instead, these factuality precisions are intended to be compared across LMs and tiers of the same evaluation methods. In Table 1, the comparison shows that (1) Harder user prompts in FactBench generally produce less factual (lower factuality precision) responses. (2) VERIFY’s ranking of LMs is more consistent than other methods across different difficulty tiers.
>
> > **2.** While FACTBENCH is described as dynamic, the paper lacks specifics on its update process, such as the conditions under which prompts are added or removed, the frequency of updates, and criteria for integrating new prompts.
>
> FACTBENCH identifies prompts within the LMSYS-chat-1M dataset that challenge LMs in factual generation. We plan to **annually** incorporate new prompts from the LMSYS-chat-1M dataset [1], which the authors intend to release quarterly. We will also expand our prompt collection by identifying hallucination prompts from the WildChat dataset [2], another rich source of user-model interactions with regular updating of the conversations. Our approach involves applying the mentioned collect and VERIFY pipeline to these new prompts, which includes extracting potential hallucination prompts, measuring their hallucination scores, comparing them against existing FACTBENCH prompts, and selecting and integrating prompts with the highest hallucination scores from the recent year. This ensures FACTBENCH remains current and diverse, captures emerging factual generation challenges, and provides a dynamic, representative benchmark for evaluating language model hallucinations.

---

> ### Author Response · Authors · 2024-11-20
> **Response to Reviewer mXTb (Part 2/2)**
>
> > **3.** The paper provides limited discussion on key hyperparameters, such as α used in the hallucination score and the number of evidence retrieval rounds in VERIFY.
>
> Below is an explanation of our design choices:
>
> **3.1.** We empirically set the evidence retrieval rounds to 5 through development testing and manual inspection of generated verification questions. This setting follows prior work [3] and proved sufficient for generating precise verification queries based on our observations. Our experiments showed that additional rounds did not meaningfully improve query quality.
>
> **3.2.** To choose the α value, we sampled 100 responses (25 per model) and asked 2 annotators to annotate whether units labeled as “undecidable” by VERIFY are supported against external evidence. Each annotator evaluates 570 units, and the inter-annotation agreement is 85.5%. The following table summarizes the portion of undecidable units that were labeled as factual vs not factual. As seen in this table, the undecidable units have a near-equal chance of being categorized as factual or not factual. Based on this finding, we set the α to 0.5 for the experiments.
> |Model|Factual (Avg. Percentage)|Not Factual (Avg. Percentage)|
> |-|-|-|
> |GPT4-o|68.4|31.6|
> |Gemini 1.5-Pro|56.6|43.4|
> |Llama 3.1-405B-Instruct|51.0|49.0|
> |Llama 3.1-70B-Instruct|52.0|48.0|
> |**Average**|**57.0**|**43.0**|
>
> **Questions**
> > **1.** What factors influenced the decision to categorize prompts into three tiers?
>
> We identify challenging prompts based on their tendency to trigger model hallucinations. However, weaker language models tend to hallucinate more frequently due to limitations in their parameter counts (affecting parametric memory and cognitive capabilities), differences in alignment techniques, and other factors. Using model performance rankings derived from pairwise human evaluations [4], this three-tier categorization offers two key advantages:
> 1. It prevents dataset bias by controlling for elevated hallucination rates from less capable models. While our dataset includes prompts queried to models of varying capabilities, we assign a higher portion of the dataset to prompts that elicit hallucinations even from strong models. This is reflected in our dataset composition: 53% Hard, 34% Moderate, and 13% Easy prompts.
> 2. As demonstrated in Section 7.1, this tiered structure enables fine-grained analysis of model performance across different difficulty levels, offering insights into both absolute capabilities and relative progress.
>
> > **2.** Why was a topic model-based approach (BERTopic) chosen over a general clustering method? ... how were these parameters tuned, and might they influence the final benchmark results?
>
> First, we want to clarify that BERTopic and general clustering methods are not mutually exclusive. BERTopic is a topic modeling pipeline based on semantic embeddings, and general clustering methods are one of the steps in the pipeline (HDBSCAN by default, but K-Means is an alternative). Unlike approaches like K-means, which require pre-specifying the number of clusters, BERTopic leverages HDBSCAN to determine the number of clusters automatically based on data density. This flexibility was essential for our user-model interaction topic analysis, where the optimal number of clusters was unknown beforehand.
>
> For parameter tuning, we used a grid search to explore combinations of key HDBSCAN parameters: `min_cluster_size` (set to 100, the minimum number of prompts per cluster) and `min_samples` (set to 25, the density threshold for outlier detection). The grid search spanned values of 10, 25, 50, 100, and 200 for both parameters. We evaluated clustering quality through manual inspection, focusing on topic granularity across the top and bottom 50 clusters while avoiding overly specific topics (e.g., "Taylor Swift's birthday") or overly general ones (e.g., "question-answering").
>
> Our manual inspection showed that clustering results remained stable across different parameter combinations, with only marginal improvements at the chosen values of `min_cluster_size=100` and `min_samples=25`. This robustness aligns with HDBSCAN's reputation for requiring minimal tuning, making it well-suited for real-world applications with limited prior knowledge of the underlying data structure. The stability of our results reinforces the reliability of our benchmark findings, minimizing the influence of parameter choices on our conclusions.
>
> **References**
> 1. Zheng, L., et al. 2024. LMSYS-Chat-1M: A Large-Scale Real-World LLM Conversation Dataset. In 2024 International Conference on Learning Representations.
> 2. Zhao, W., et al. 2024. WildChat: 1M ChatGPT Interaction Logs in the Wild. In 2024 International Conference on Learning Representations.
> 3. Wei, J., et al. 2024. Long-form factuality in large language models. In 2024 Conference on Neural Information Processing Systems.
> 4. Arena Leaderboard. https://huggingface.co/spaces/lmsys/chatbot-arena-leaderboard

---

> > ### Comment · Reviewer_mXTb · 2024-11-27
> >
> > Thank you for the detailed response and updates addressing my feedback. I appreciate the clarification regarding inter-annotator agreement, which was well-explained and has resolved some of my concerns.
> >
> > However, I remain concerned about the dynamic update process for FACTBENCH. While the planned integration of new prompts from LMSYS-chat-1M and WildChat datasets is described, the overall update process lacks sufficient detail. For instance, it is unclear how new prompts are evaluated over time to determine their inclusion, what specific criteria and thresholds are used for selecting high hallucination score prompts, and how these criteria and thresholds will be adjusted as the benchmark evolves to maintain its representativeness.
> > Similarly, although the justification for setting α=0.5 is reasonable, its fixed nature may limit adaptability across different tasks or models. Addressing these points in more detail could further enhance the clarity and robustness of the paper.

---

> > > ### Author Response · Authors · 2024-11-28
> > > **Response to Reviewer mXTb**
> > >
> > > Thanks for your response. We would be happy to share more details to address your concerns:
> > > >The Dynamic Update Process for FACTBENCH
> > >
> > > We appreciate your concerns about the FACTBENCH update process. Let us clarify our comprehensive approach:
> > >
> > > Upon receiving new wild interaction data for a new year, we will directly apply the existing Data Cleaning and Data Clustering pipelines to identify the most representative prompt clusters from that year's data. Subsequently, we will evaluate these prompts for Verifiability and Usefulness using the same parameters and pipelines as before. From this, we can obtain a series of new prompts.
> > >
> > > After this step, we need to combine the resulting prompts with the existing FactBench prompts. At this stage, we consider two challenges:
> > > 1. Old prompts may overlap with the new prompts. To address this issue, we check whether each old prompt belongs to the clusters fitted by the new prompts and remove those with excessive similarity.
> > >
> > > 2. Some proprietary models are continually updated, meaning some old prompts may no longer be challenging. Therefore, we will regenerate responses to the old prompts on the latest versions of the models.
> > >
> > > After addressing these two challenges, we obtain a combined prompt set with the latest model responses. From this, we can run our VERIFY pipeline to compute hallucination scores. Based on the score rankings, we select a new set of prompts to form the new version of FACTBENCH.
> > >
> > > > Regarding fixed α Parameter across tasks and models
> > >
> > > We recognize your point about α's fixed nature. However, α selection heavily relies on the model's internal knowledge. For any new model or task, determining the optimal α requires human annotation, which we believe contradicts the original purpose of automating the factuality evaluation. Results of per-model annotation show similar α values, suggesting a value of 0.5 is a reasonable estimation across models and balancing cost-effectiveness and performance.
> > >
> > > We will incorporate these detailed explanations in our final paper revision and would appreciate it if you could consider the provided context in evaluating our work. Please let us know if you need further clarification on these or other aspects of our methodology.

---

### Official Review · Reviewer_xqv7 · 2024-11-04

**Soundness:** 3
**Presentation:** 3
**Contribution:** 2
**Rating:** 5
**Confidence:** 4

**Summary:**

This paper presents FACTBENCH, a dynamic benchmark dataset for evaluating the factuality of language model (LM) responses in real-world user interactions. The authors introduce a two-step process to curate the benchmark: (1) collecting verifiable and useful prompts from an in-the-wild LM conversation dataset, and (2) using VERIFY, a factuality evaluation pipeline, to measure the appropriateness of these prompts based on whether they elicit unfactual responses from strong LMs. The resulting FACTBENCH contains 985 hallucination prompts across 213 topics. The authors also benchmark several widely-used LMs on FACTBENCH and find that proprietary models exhibit better factuality than open-weight models, and VERIFY achieves the highest correlation with human judgments compared to other factuality evaluation methods.

**Strengths:**

1. FACTBENCH is a novel dynamic benchmark that captures evolving factuality challenges in real-world LM interactions, addressing the limitations of existing static benchmarks.

2. VERIFY, the factuality evaluation pipeline, considers the verifiability of generated content and introduces an "undecidable" label for ambiguous cases, providing a more robust framework for assessing factuality.

3. The authors release human-annotated factuality data on 5,519 content units, which can serve as a valuable resource for future research on factuality evaluation.

**Weaknesses:**

1. The usefulness evaluation criteria and scoring process are not well-justified. More details are needed on how these criteria were determined and how the scores from two LMs were combined.

2. The VERIFY pipeline uses a single LM (Llama3-70B-Instruct) for key tasks such as unit extraction, labeling, and decontextualization. The potential bias introduced by relying on a single model is not adequately addressed.

3. The evaluation of FACTBENCH is limited to a small set of LMs. A more comprehensive evaluation involving a wider range of models would strengthen the paper's claims.

4. The weighting factor α in the Hallucination Score is set to 0.5 without much explanation. A sensitivity analysis or ablation study on this hyperparameter would be informative.

**Questions:**

N/A

---

> ### Author Response · Authors · 2024-11-19
> **Response to Reviewer xqv7 (Part 1/2)**
>
> Thank you for your thorough and insightful review of our paper. We appreciate your detailed feedback and your constructive suggestions. Below, we address your concerns:
>
> > **1**. The usefulness evaluation criteria and scoring process are not well-justified. More details are needed on how these criteria were determined and how the scores from two LMs were combined.
>
> As described at the end of Section 3, instead of randomly selecting prompts from the collection of cleaned, diverse, and verifiable prompts, we first assign each prompt a “usefulness” score and then select the highest scoring prompts for automated fact-checking. Through multiple rounds of discussion and empirical testing, we refined the criteria to capture the most critical aspects of prompt usefulness:
> - Clarity: This criterion assesses whether the prompt is easily understandable and is not ambiguous.
> - Generalizability: We developed this criterion to prevent over-specialization. The assessment focuses on the prompt's potential to be meaningful across different contexts or users.
> - Relevance: This criterion assesses whether the information requested is important and potentially interesting to a broader audience.
> - Feasibility: This criterion evaluates whether the requested information is reasonably provided within the LM's capabilities.
>
> Our scoring methodology involved two frontier LMs (GPT-4 Turbo and Llama3-70B-Instruct) independently scoring each criterion on a scale from 0 (lowest) to 5 (highest). The aggregate score calculation leverages a formula that balances multiple models' perspectives:
>
> $$
> S(P) = \frac{1}{|C|} \sum_{c \in C} \sum_{m \in M} S_m(c)
> $$
>
> Where $ C $ denotes the set of criterion $ C $ = {clarity, generalizability, relevance, feasibility}, $ M $ denotes the set of models (GPT4-Turbo, Llama3-70B-instruct), and $ S_m(c) $ denotes the score that model $ m $ assigns to criteria $ c $.
>
> This approach reduces individual model bias and ensures a comprehensive evaluation of prompt usefulness, allowing us to create a more robust and reliable dataset for further research and analysis.
>
> > **2**. The VERIFY pipeline uses a single LM (Llama3-70B-Instruct) for key tasks such as unit extraction, labeling, and decontextualization. The potential bias introduced by relying on a single model is not adequately addressed.
>
> Our pipeline follows a similar pattern in previous factuality evaluation work to use a single LM as the backbone LM of the pipeline [1, 2, 3]. Alternatively, our pipeline is model-agnostic and can be instantiated with different LMs of the user’s choice. In this work, Llama3-70B-Instruct was selected as it was the best open-weight LM in terms of cost, efficiency, and reproducibility, especially considering the limitations of proprietary models.
>
> We acknowledge the potential of using multiple LMs to enhance diversity and reduce bias. However, critical challenges remain, such as determining optimal strategies for merging unit extraction and decontextualization across models. Key research questions include whether to use a single model for initial unit extraction/decontextualization stages and then collect verification labels from multiple models, how to implement model collaboration—whether through simple ensembling methods or more sophisticated collaborative decision-making—and how to maintain computational efficiency throughout the evaluation process. These methodological and computational challenges require careful investigation to develop a robust multi-model factuality evaluation approach.
>
>
> **References**
>
> 1. Min, S., et al. 2023. FActScore: Fine-grained atomic evaluation of factual precision in long-form text generation. In Proceedings of the 2023 Conference on Empirical Methods in Natural Language Processing.
>
> 2. Wei, J., et al. 2024. Long-form factuality in large language models. In 2024 Conference on Neural Information Processing Systems.
>
> 3. Wang, Y., et al. 2024. Factcheck-bench: Fine-grained evaluation benchmark for automatic fact-checkers. In Findings of the Association for Computational Linguistics: EMNLP 2024.

---

> ### Author Response · Authors · 2024-11-19
> **Response to Reviewer xqv7 (Part 2/2)**
>
> > **3.** The evaluation of FACTBENCH is limited to a small set of LMs. A more comprehensive evaluation involving a wider range of models would strengthen the paper's claims.
>
> We acknowledge your concern. Due to time and resource constraints, we conducted an in-depth analysis of the four most representative and popular LMs at the time of submission. However, as mentioned in our paper, FactBench is a continuously updated benchmark, with new LMs and user prompts consistently added when they are out. Below is our up-to-date FactBench Leaderboard (1K questions) with 3 newly added LMs (CommandR+, Mistral-Large-2, Claude-3.5-Sonnet):
> |Tier|Rank|Model|Factual Precision|Hallucination Score|Avg. # Tokens|Avg. # Units|Avg. # Undecidable|Avg. # Unsupported|
> |----|----|-----|-----------------|-------------------|-------------|------------|------------------|------------------|
> |**Tier 1: Hard**|1|GPT4-o|75.65|0.64|563.15|24.01|4.62|1.01|
> | |2|Mistral-Large-2|75.19|0.67|485.58|23.21|4.09|1.36|
> | |3|Claude-3.5-Sonnet|74.95|0.65|395.77|22.64|4.03|1.19|
> | |4|Gemini1.5-Pro|73.78|0.68|517.31|22.25|4.48|1.13|
> | |5|CommandR+|73.15|0.71|440.93|23.55|4.51|1.40|
> | |6|Llama3.1-70B-Instruct|70.07|0.89|532.41|27.17|5.67|2.13|
> | |7|Llama3.1-405B-Instruct|68.59|0.93|551.28|26.71|6.19|2.20|
> |**Tier 2: Moderate**|1|GPT4-o|80.72|0.50|624.67|24.42|3.59|0.89|
> | |2|CommandR+|80.71|0.52|483.32|24.10|3.17|1.09|
> | |3|Mistral-Large-2|79.97|0.52|528.44|22.65|3.21|1.02|
> | |4|Claude-3.5-Sonnet|79.92|0.54|414.32|22.15|3.32|1.09|
> | |5|Gemini1.5-Pro|78.02|0.57|565.97|22.16|3.71|0.97|
> | |6|Llama3.1-70B-Instruct|75.76|0.71|607.44|25.35|4.33|1.76|
> | |7|Llama3.1-405B-Instruct|75.05|0.70|599.30|25.24|4.74|1.41|
> |**Tier 3: Easy**|1|Mistral-Large-2|92.00|0.25|523.57|27.80|1.80|0.55|
> | |2|CommandR+|91.65|0.25|499.06|27.95|1.57|0.54|
> | |3|GPT4-o|91.63|0.26|640.84|29.29|2.01|0.53|
> | |4|Gemini1.5-Pro|89.86|0.31|551.81|25.60|1.88|0.71|
> | |5|Claude-3.5-Sonnet|89.61|0.30|411.20|26.72|1.49|0.81|
> | |6|Llama3.1-70B-Instruct|89.30|0.33|607.75|31.38|2.08|0.83|
> | |7|Llama3.1-405B-Instruct|86.57|0.40|599.87|30.12|2.88|0.85|
>
> > **4.** The weighting factor α in the Hallucination Score is set to 0.5 without much explanation. A sensitivity analysis or ablation study on this hyperparameter would be informative.
>
> Thanks for pointing this out. The weighting factor α balances the importance of undecidable and unsupported units. It accounts for LMs generating content from training data, which may include factual information not found in current Web searches. To choose the α value, we sampled 100 responses (25 per model) and asked 2 annotators to annotate whether units labeled as “undecidable” by VERIFY are supported against external evidence. Each annotator evaluates 570 units, and the inter-annotation agreement is 85.5%. The following table summarizes the portion of undecidable units that were labeled as factual vs not factual. As seen in this table, the undecidable units have a near-equal chance of being categorized as factual or not factual. Based on this finding, we set the α to 0.5 for the experiments.
>
> | Model                          | Factual (Avg. Percentage) | Not Factual (Avg. Percentage) |
> |--------------------------------|----------------------------|-------------------------------|
> | GPT4-o                         | 68.4                       | 31.6                          |
> | Gemini 1.5-Pro                 | 56.6                       | 43.4                          |
> | Llama 3.1-405B-Instruct        | 51.0                       | 49.0                          |
> | Llama 3.1-70B-Instruct         | 52.0                       | 48.0                          |
> | **Average**                    | **57.0**                       | **43.0**                          |

---

### Comment · Area_Chair_Y5es · 2024-11-25

Dear reviewers,

As the deadline for discussion is ending soon. Please respond to the authors to indicate you have read their rebuttal. If you have more questions, now is the time to ask. This is important since the paper is currently undergoing extremely divergent scores.

AC

---

### Author Response · Authors · 2024-12-02
**Request for further feedbacks from reviewers**

Dear reviewers,

We thank you again for your valuable comments. We have provided detailed clarifications and supplementary experiments to address your concerns and will include them in our final version.
Thank you for your help in improving our work. If our rebuttal satisfactorily addressed your concerns, we kindly request that you reconsider your assessments.

Authors of Submission13591

---

### Note · Authors · 2025-02-16

I have read and agree with the venue's withdrawal policy on behalf of myself and my co-authors.

---

### Meta-Review · Area_Chair_Y5es · 2024-12-18

**Metareview:**

This paper introduces FACTBENCH, a dynamic benchmark aimed at evaluating the factuality of language model responses by identifying and categorizing hallucinations. The research addresses an important challenge in LLM development by proposing VERIFY, a pipeline that assesses response factuality across 985 prompts spanning 213 topics. Key strengths include its interesting approach to capturing real-world interaction challenges, the introduction of an "undecidable" label for ambiguous cases, and the potential to provide a valuable resource for future factuality research. The authors demonstrate that proprietary models generally perform better in factuality tests, and their methodology shows promising alignment with human judgments across multiple evaluation dimensions.

However, the paper suffers from significant limitations that finally lead to its rejection. Reviewers consistently pointed out critical weaknesses, including a poorly justified usefulness evaluation process, over-reliance on a single model (Llama3-70B-Instruct) for key tasks, which introduces potential bias (xqv7, mXTb). The experimental design shows limited comprehensiveness, with a small set of evaluated LMs and unconvincing difficulty ratings (BPdM, 42L8). Methodological concerns are further compounded by low inter-annotator agreement, with Cohen's Kappa scores of only 0.52 and 0.55, raising questions about annotation reliability (mXTb). Additionally, the paper lacks transparency in critical areas such as the update process for the benchmark, justification for key hyperparameters (particularly the weighting factor \alpha), and a comprehensive analysis of the methodology's limitations (xqv7, mXTb, BPdM). These shortcomings, coupled with the incremental nature of the contribution, led reviewers to rate the paper marginally below the acceptance threshold.

Overall, this is a good paper. But it should be significantly polished before accepted.

**Additional Comments On Reviewer Discussion:**

Reviewers are not satisfied with the response towards limited experimental results, limited models, and application scenarios.

---

### Decision · Program_Chairs · 2025-01-22

Reject